

# $M_k$ models: the field theory connection

**Thessa Fokkema and Kareljan Schoutens**[⋆]

Institute for Theoretical Physics Amsterdam and Delta Institute for Theoretical Physics,
University of Amsterdam, Science Park 904, 1098 XH Amsterdam, The Netherlands.

⋆ C.J.M.Schoutens@uva.nl

## Abstract

The $M_k$ models for 1D lattice fermions are characterised by $\mathcal{N} = 2$ supersymmetry and by an order-$k$ clustering property. This paper highlights connections with quantum field theories (QFTs) in various regimes. At criticality the QFTs are minimal models of $\mathcal{N} = 2$ supersymmetric conformal field theory (CFT) - we analyse finite size spectra on open chains with a variety of supersymmetry preserving boundary conditions. Specific staggering perturbations lead to a gapped regime corresponding to massive $\mathcal{N} = 2$ supersymmetric QFT with Chebyshev superpotentials. At 'extreme staggering' we uncover a simple physical picture with degenerate supersymmetric vacua and mobile kinks. We connect this kink-picture to the Chebyshev QFTs and use it to derive novel CFT character formulas. For clarity the focus in this paper is on the simplest models, $M_1$, $M_2$ and $M_3$.

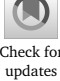
## Contents

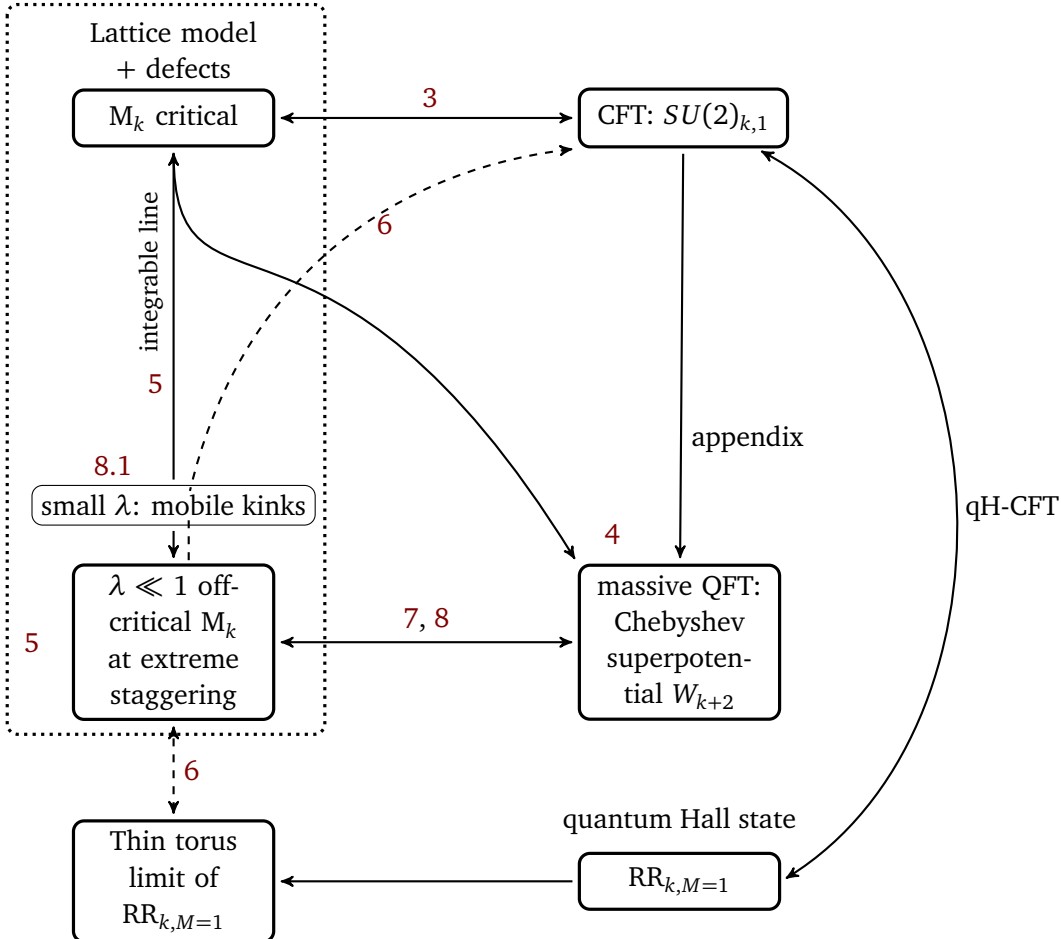

Figure 1: Overview of the connections studied in this paper. The numerals going with the arrows indicate the section where the relation is discussed. Some of the important connections are as follows. (i) The critical $M_k$ model is related to the $k$-th superconformal $\mathcal{N} = 2$ minimal model, here denoted as $SU(2)_{k,1}$ (section 3). (ii) The massive QFT with Chebyshev superpotentials arises from the weakly staggered lattice model through RG flow, or alternatively from a relevant perturbation of the CFT (section 4, appendix). (iii) The off-critical lattice model can be studied in the extreme staggering limit $\lambda \ll 1$ and, close to this limit, in perturbation theory in $\lambda$ (section 5). A direct relation between the extreme staggering limit and the CFT can be made by counting the kinks in this limit and relating the counting formulas to characters in the CFT (section 6). (iv) The $k$-th minimal CFT is via the quantum Hall-CFT correspondence related to the Read-Rezayi quantum Hall state with order-$k$ clustering, here denoted as $RR_k$ (section 6). (v) The lattice model kinks at extreme staggering are in many ways similar to the fundamental kinks in the massive QFT - we compare the two in sections 7, 8.

## 1  Introduction

Field theory connections are an important and universal element of the toolbox that theoretical physicists employ in their analysis of lattice models for strongly correlated quantum materials. General arguments based on the renormalisation group link critical phases to quantum field theories with a scaling or conformal symmetry, while gapped phases correspond to massive quantum field theories. In general, it is a highly non-trivial task to set up a dictionary between, on the one hand, parameters and observables in the lattice model and, on the other, couplings and field operators in the quantum field theory (QFT). In making these connections,

the fundamental symmetries of the microscopic lattice model are a strong guiding principle. Symmetries of the microscopic model have counterparts in the QFT description. In addition, the QFT typically displays additional symmetries that are absent in the microscopic model but emerge in the RG flow. An example of the latter are the (infinite-dimensional) conformal symmetries in the continuum description of critical lattice models in one spatial dimension.

In this paper we report on field theory connections for a particular class of lattice models, the so-called supersymmetric $M_k$ models in one spatial dimension (see section 2 for a concise introduction). These models possess an explicit $\mathcal{N} = 2$ supersymmetry on the lattice, which connects to various notions of $\mathcal{N} = 2$ space-time supersymmetry in the corresponding QFTs. We zoom in on special choices of $M_k$ model parameters, which are such that these models are integrable, in the sense of admitting a solution by Bethe Ansatz. The corresponding QFTs are then integrable as well - in particular, the massive QFTs describing the gapped phases admit a description in terms of particles with factorisable scattering matrix.

The combination of supersymmetry and integrability turns out to be particularly potent in structuring both lattice models and quantum field theories, as has been known and exploited in many settings. In the specific context of the $M_k$ lattice models, some striking results have been reported in the literature. The critical $M_k$ model corresponds to the $k$-th minimal model of $\mathcal{N} = 2$ superconformal field theory [1], and there is a precise understanding of how special (so-called $\sigma$-type) boundary conditions on the critical $M_k$ chains translate into CFT boundary fields and open chain CFT partition sums [2,3]. Specific integrable deformations of the critical models, obtained by staggering some of the couplings, connect to a specific class of integrable $\mathcal{N} = 2$ supersymmetric QFT, characterised, in their superfield formulation, by superpotentials taking the form of so-called Chebyshev polynomials $W_{k+2}$ [2,4]. These connections, which we review in sections 3, 4, constitute the beginnings of a detailed understanding of the lattice model-to-field theory dictionary for the $M_k$ models.

In this paper we report on further $M_k$ model-to-field theory connections. These have their origin in a simple physical picture that arises if we follow the deformed critical models into the regime of what we call 'extreme staggering' (section 5). In this regime, a simple physical picture emerges, based on $k + 1$ degenerate ground states with a simple, tractable form and excitations that take the shape of kinks connecting these various vacua. These kinks satisfy specific exclusion statistics rules. At strong but finite staggering the kinks become mobile, giving a spectrum that is easily understood in terms of a (non-relativistic) band structure. Changing the strength of the staggering deformation gives a continuous interpolation between this simple 'mobile kink' picture and the $M_k$ model at criticality. In section 6 we employ this connection to obtain expressions for CFT characters as $q$-deformations of characters that describe the kink spectrum at extreme staggering. In doing this analysis, we used the fact that the systematics of the kinks at extreme staggering are in many ways analogous to those of quasi-hole excitations over the $k$-clustered Read-Rezayi quantum Hall states [5,6] in the so-called thin-torus limit [7–9].

The kink picture at extreme staggering is remarkably close to the physical picture arising from the particle description of the QFTs that constitute the RG fixed points of the weakly staggered $M_k$ models. At the same time, the kinematical settings are very different: the kinks at extreme staggering have a non-relativistic band structure while the QFT kinks are fully relativistic. Both regimes enjoy a high degree of supersymmetry (for example: we will see that the $M_2$ model admits a total of six supercharges in both regimes), but since these supercharges anti-commute into operators for momentum and energy, their action on kink states is necessarily very different between the two regimes. In section 7 we analyse this situation in some detail for the $M_2$ model.

In section 8 we make a further comparison between the kink picture at strong staggering and the particle picture of the relativistic QFT. Concentrating on the $M_2$ model, we focus

on the effect of non-diagonal boundary scattering induced by non-trivial ($\sigma$-type) boundary conditions on an open chain. We compare the result of a perturbative calculation at strong staggering with a QFT computation having for input a non-diagonal boundary reflection matrix.

## 2  $M_k$ models: definitions and basic properties

The $M_k$ models, first introduced in [1,10], are lattice models of interacting particles with an explicit $\mathcal{N} = 2$ supersymmetry. The particles on the lattice are fermions without spin. The models can be defined on general graphs but we will only consider the model defined on a one-dimensional open or closed chain of length $L$. In the $M_k$ model the spinless fermions are subject to an exclusion rule which allows a group of at most $k$ fermions on neighbouring sites:

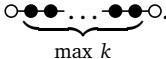

The Hamiltonian of the model is defined in terms of fermion creation and annihilation operators via the supercharges. The supercharge $Q_+$ decreases the fermion number $f \to f - 1$ and its hermitian conjugate $Q_+^\dagger = \bar{Q}_+$ increases the fermion number $f \to f + 1$. The operator $\bar{Q}_+$ is written in terms of constrained fermionic creation operators $d^\dagger_{[a,b],j}$ which create a particle at lattice site $j$ in such a way that a string of $a$ particles is formed, with the newly created particle at the $b$-th position in the string, $1 \le b \le a$. This process has an amplitude given by $\lambda_{[a,b],j}$,

$$\bar{Q}_+ = \sum_{j=1}^{L} \sum_{a,b} \lambda_{[a,b],j} d^\dagger_{[a,b],j} \qquad \overset{\overset{b}{\downarrow}}{\circ\bullet\bullet\underbrace{\bullet\textcolor{red}{\bullet}\bullet\bullet}_{a}\bullet\circ}, \tag{1}$$

where the sum is over the sites $j$ on the lattice. The operators $d^\dagger_{[a,b],j}$ can be written in terms of the usual fermion creation and annihilation operators $c_j, c_j^\dagger$ which satisfy $\{c_i, c_j^\dagger\} = \delta_{i,j}$ and $\{c_i, c_j\} = \{c_i^\dagger, c_j^\dagger\} = 0$. For this we use the projection operator $\mathscr{P}_j = 1 - c_j^\dagger c_j$. For the $M_1$ model we only need the constrained fermion creation operator $d^\dagger_{[1,1],j}$ which is given by

$$d^\dagger_{[1,1],j} = \mathscr{P}_{j-1} c_j^\dagger \mathscr{P}_{j+1}. \tag{2}$$

For the $M_2$ model also $d^\dagger_{[2,1],j}$ and $d^\dagger_{[2,2],j}$ are needed and they are given by

$$d^\dagger_{[2,1],j} = \mathscr{P}_{j-1} c_j^\dagger c_{j+1}^\dagger c_{j+1} \mathscr{P}_{j+2}, \quad d^\dagger_{[2,2],j} = \mathscr{P}_{j-2} c_{j-1}^\dagger c_{j-1} c_j^\dagger \mathscr{P}_{j+1}. \tag{3}$$

Similarly all $d^\dagger_{[a,b],j}$ are defined for the $M_k$ models. For $Q_+$ and $\bar{Q}_+$ to be true supercharges, we require that

$$(Q_+)^2 = 0, \quad (\bar{Q}_+)^2 = 0. \tag{4}$$

This property does not hold for general values of the parameters $\lambda_{[a,b],j}$. Below we address the freedom we have in the choice of parameters.

The Hamiltonian of each of the $M_k$ models is now defined as the anti-commutator of the nilpotent supercharges $Q_+$ and $\bar{Q}_+$:

$$H = \{Q_+, \bar{Q}_+\}. \tag{5}$$

By construction, $H$ commutes with both supercharges $Q_+$ and $\bar{Q}_+$. Although $Q_+$ and $\bar{Q}_+$ are nonlocal, taking their anti-commutator leads to a local Hamiltonian with an interaction range of a maximum of $k$ sites.

The $M_k$ models were first introduced with the parameters $\lambda_{[a,b],j} = \lambda_{[a,b]}$, thus independent of the lattice site $j$. In [11] staggering was introduced for the $M_1$ model (it was further studied in [12–14]), and in ref. [4] the staggered $M_2$ model has been considered. In the case the amplitudes do not depend on the site $j$, we call the model homogeneous. In the case where the amplitudes $\lambda_{[a,b],j}$ have an explicit site dependence we say that the amplitudes are staggered and we call the model inhomogeneous.

The restriction $(Q_+)^2 = 0$ gives relations on the coefficients $\lambda_{[a,b],j}$, reducing the number of free parameters. This restriction is equivalent to equating the amplitudes of two processes: one in which from a string of length $a$ the particle at position $b$ and then the particle at position $c$ ($b < c$) are removed, and the other in which these particles are annihilated in the opposite order. The particle at position $c$ becomes a particle at position $c - b$ of a string of length $a - b$ after a particle at position $b$ has been removed. This leads to the recursion relation:

$$\lambda_{[a,b],j}\lambda_{[a-b,c-b],j+c-b} = \lambda_{[a,c],j+c-b}\lambda_{[c-1,b],j} \qquad 1 \le b < c \le a. \tag{6}$$

This can be solved by [1, 15]

$$\lambda_{[a,b],j} = \left(\prod_{k=1}^{b-1} \frac{\lambda_{[a-k+1,1],j-b+k}}{\lambda_{[b-k,1],j-b+k}}\right)\lambda_{[a-b+1,1],j}. \tag{7}$$

In the homogeneous case, $\lambda_{[a,b],j} = \lambda_{[a,b]}$, this gives

$$\lambda_{[a,b]} = \frac{\lambda_{[a,1]}\lambda_{[a-1,1]}\cdots\lambda_{[a-b+1,1]}}{\lambda_{[b-1,1]}\lambda_{[b-2,1]}\cdots\lambda_{[1,1]}} \tag{8}$$

so only $\lambda_{[1,1]}, \lambda_{[2,1]}, \ldots, \lambda_{[k,1]}$ are left as free parameters. Since we can choose a normalisation of the Hamiltonian one of these parameters can be set to 1, which gives a total of $k - 1$ free parameters for the homogeneous $M_k$ model.

The paper [15] obtained a 1-parameter family of couplings $\lambda_{[a,b],j}$ which describe a supersymmetric, integrable staggering perturbation of the critical point of the homogeneous $M_k$ model. These staggerings are periodic with a period of $k + 2$ lattice sites. Our choice of couplings for $k = 1, 2, 3$, which we describe below, agree with this choice of parameters.

## 2.1 $M_1$ model

The $M_1$ model is integrable and critical in the homogeneous case, where $\lambda_{[1,1],j} = 1$ for all $j$. Criticality is lost when the couplings are staggered; however, it was found that the $M_1$ model is integrable for all types of staggering modulo 3 [19]. In this paper we focus on the staggering pattern

$$\lambda_{[1,1],j}: \quad \ldots \quad \lambda \quad 1 \quad 1 \quad \lambda \quad 1 \quad 1 \quad \ldots \tag{9}$$

which we denote by $\ldots \lambda 11\lambda 11 \ldots$.

## 2.2 $M_2$ model

For the $M_2$ model eq. (6) gives

$$\lambda_{[2,1],j-1}\lambda_{[1,1],j} = \lambda_{[2,2],j}\lambda_{[1,1],j-1} \tag{10}$$

which leads to the parametrisation

$$\lambda_{[1,1],j} = \sqrt{2}\lambda_j, \quad \lambda_{[2,1],j} = \sqrt{2}\lambda_j\mu_j, \quad \lambda_{[2,2],j} = \sqrt{2}\lambda_j\mu_{j-1}. \tag{11}$$

It follows that in the homogeneous case $\lambda_{[2,1]} = \lambda_{[2,2]}$, so in this case there is a symmetry between annihilating the first and the second particle of a pair of two particles. If we want this property also in the staggered case we have to set $\mu_j = \mu$ for all $j$.

In this paper we put $\mu_j = \mu_{j+1} = 1/\sqrt{2}$ and focus on the staggering pattern

$$
\begin{aligned}
\lambda_{[1,1],j} &: \quad \ldots \quad \sqrt{2} \quad \sqrt{2}\lambda \quad \sqrt{2} \quad \sqrt{2}\lambda \quad \sqrt{2} \quad \ldots \\
\lambda_{[2,1],j} &: \quad \ldots \quad 1 \quad \lambda \quad 1 \quad \lambda \quad 1 \quad \ldots \\
\lambda_{[2,1],j} &: \quad \ldots \quad 1 \quad \lambda \quad 1 \quad \lambda \quad 1 \quad \ldots
\end{aligned} \tag{12}
$$

which we denote by by $\ldots 1\lambda1\lambda1\ldots$. For this staggering the $M_2$ model Hamiltonian simplifies. The potential terms give 2 or $2\lambda^2$ (depending on the site) for creating or annihilating an isolated particle and 1 or $\lambda^2$ for creating or annihilating a particle that is part of a pair. The kinetic terms are

$$
\begin{aligned}
& \text{with amplitude} \quad \lambda, \\
& \text{with amplitude} \quad -\lambda^2 \text{ or } -1, \\
& \text{with amplitude} \quad \sqrt{2}\lambda, \\
& \text{with amplitude} \quad \sqrt{2}\lambda, \\
& \text{with amplitude} \quad \lambda.
\end{aligned} \tag{13}
$$

For the second process the value depends on the site $j$.

For $\lambda = 1$ the $M_2$ model is critical. A deformation where $\lambda < 1$ gives an RG flow to a supersymmetric sine-Gordon theory (see section 4). In section 5 we study the $M_2$ model in the limit of extreme staggering.

## 2.3 $M_3$ model

For the $M_3$ model eq. (6) gives

$$
\begin{aligned}
\lambda_{[3,2],j}\lambda_{[1,1],j-1} &= \lambda_{[2,1],j}\lambda_{[3,1],j-1} \\
\lambda_{[3,3],j}\lambda_{[1,1],j-1}\lambda_{[2,1],j-2} &= \lambda_{[1,1],j}\lambda_{[2,1],j-1}\lambda_{[3,1],j-2}.
\end{aligned} \tag{14}
$$

We can add to the parametrisation of eq. (11) the following relations to satisfy $(Q_+)^2 = 0$ for the $M_3$ model:

$$
\lambda_{[3,1],j} = \lambda_j \mu_j \nu_j, \quad \lambda_{[3,2],j} = \lambda_j \mu_j \mu_{j-1} \nu_{j-1}, \quad \lambda_{[3,3],j} = \lambda_j \mu_{j-1} \nu_{j-2}. \tag{15}
$$

In this paper we assume a particular choice of couplings, obtained in [15], which describe a critical point perturbed by an integrable staggering perturbation with lattice periodicity 5. At the critical point the couplings are

$$
\lambda_{[1,1],j} = y, \quad \lambda_{[2,1],j} = \lambda_{[2,2],j} = 1, \quad \lambda_{[3,1],j} = \lambda_{[3,3],j} = y, \quad \lambda_{[3,2],j} = 1/y^2
$$

with $y = \sqrt{\dfrac{1+\sqrt{5}}{2}}$. \tag{16}

At extreme staggering, $\lambda \ll 1$, the couplings read, to lowest order in $\lambda$,

$$
\begin{array}{llllllll}
\lambda_{[1,1],j}: & \dots & 1 & \sqrt{2} & \sqrt{2}\lambda & \sqrt{2} & 1 & \dots \\
\lambda_{[2,1],j}: & \dots & 1 & 1 & \lambda & \sqrt{2} & \lambda & \dots \\
\lambda_{[2,2],j}: & \dots & \lambda & \sqrt{2} & \lambda & 1 & 1 & \dots \\
\lambda_{[3,1],j}: & \dots & 1 & \lambda & \lambda & 1 & \lambda/\sqrt{2} & \dots \\
\lambda_{[3,2],j}: & \dots & \lambda/\sqrt{2} & 1 & \lambda^2/\sqrt{2} & 1 & \lambda/\sqrt{2} & \dots \\
\lambda_{[3,3],j}: & \dots & \lambda/\sqrt{2} & 1 & \lambda & \lambda & 1 & \dots
\end{array}
\tag{17}
$$

where the dots indicate repetition modulo 5. We denote this as $\dots \star \star \lambda \star \star \dots$, with the '$\lambda$' indicating the central position in the staggering pattern.

## 3   CFT description of critical $M_k$ models

The critical $M_k$ model corresponds to the $k$-th minimal model of $\mathcal{N} = 2$ CFT [1]. In this section we demonstrate how this correspondence works out for $M_k$ model spectra on open chains. The main finding, which we briefly reported in [2], is a precise map between a choice of boundary conditions on the chain and the CFT modules describing the open chain spectra.

Throughout this paper we use a description where the $k$-th minimal model of $\mathcal{N} = 2$ supersymmetric CFT is represented as a product of a free boson CFT times a $\mathbb{Z}_k$ parafermion theory. For $k = 2$ the parafermion fields are a Majorana fermion $\psi$ and a spin field $\sigma$, while for general $k > 1$ we have parafermions $\psi_1, \dots, \psi_{k-1}$ together with a collection of spin fields. A typical operator in the supersymmetric CFT has a factor originating in the parafermion theory and a factor from the free boson part, the latter taking the form of a vertex operator $V_{p,q}$. Other than in a stand-alone free boson theory, not all charges $p, q$ are integers. The $\mathcal{N} = 2$ supercurrents are represented as

$$
G_L^{k,+} = \psi_1 V_{1,\frac{k+2}{2k}}, \quad G_L^{k,-} = \psi_{k-1} V_{-1,-\frac{k+2}{2k}}, \quad G_R^{k,+} = \overline{\psi}_1 V_{1,-\frac{k+2}{2k}}, \quad G_R^{k,-} = \overline{\psi}_{k-1} V_{-1,\frac{k+2}{2k}}.
\tag{18}
$$

### 3.1   $M_1$ spectra

In [16, 17] it was established that finite size spectra for the $M_1$ model on open chains correspond to irreducible modules of the first minimal model of $\mathcal{N} = 2$ supersymmetric CFT. Their highest weight states are created by chiral vertex operators of charge $m$; we use the notation $V_m$ to denote both these vertex operators and the corresponding modules. Depending on $L \mod 3$, all Ramond sector modules of the supersymmetric CFT are realised by the $M_1$ model with open boundary conditions. [We remark that Neveu-Schwarz sectors are not compatible with lattice supersymmetry. The Neveu-Schwarz vacuum sector in particular gives $E_0 = -\frac{c}{24} < 0$, whereas $E \geq 0$ for all states in the supersymmetric lattice model.] For higher $k$, the complete lattice model-to-CFT correspondence requires more general supersymmetric boundary conditions, called $\sigma$-type BC, which were first introduced in [2, 3].

### 3.2   $M_2$ model

For the $M_2$ model $\sigma$-type BC arise if we impose the constraint that the two sites adjacent to a boundary cannot both be occupied by a particle ('no 11'). Another implementation of this constraint is forbidding the site at the boundary to be empty ('no 0') which has as an effect that the two sites adjacent to it cannot both be occupied by a particle. A chain of length $L$ with the 'no 11' condition at the boundary is thus similar to a chain of length $L + 1$ with the

'no 0' condition at the boundary. The only difference is a relative factor of $\sqrt{2}$ for creating or annihilating a particle on the site that is at the boundary in the former description and second from the boundary in the latter. At extreme staggering this difference is important, it can change the number of elementary kinks. However, we expect that at criticality this difference corresponds to an irrelevant perturbation of the conformal field theory.

The numerical finite size spectra for the critical $M_2$ model with open/open, $\sigma$/open and $\sigma/\sigma$ BC can be matched to (combinations of) CFT modules $V_m$, $\sigma V_m$ and $\psi V_m$. In the correspondence, a $\sigma$-type BC corresponds to acting on the CFT modules with the operator $\sigma V_{1/2}$. We briefly summarise these results, which we established in our paper [2], in the next section.

### 3.2.1 $M_2$ finite size spectra and CFT characters

The CFT finite size spectra in the Ramond sector are built by acting with the modes of $\partial \varphi$ and of $\psi$ on the highest weight states $\sigma V_m$ ($m$ integer) and $V_m$ ($m$ half-integer). On the first type, with $m$ integer, the $\psi$ modes are $\psi_{-l}$, $l = 1, 2, \ldots$. On the second type, with $m$ half-integer, the $\psi$ modes are $\psi_{-l+1/2}$, $l = 1, 2, \ldots$. The character formulas for the fermion part of the CFT are given by

$$\text{ch}(q) = \sum_n \frac{q^{\frac{1}{2}n^2 + an + b}}{(q)_n} \,, \tag{19}$$

with $(q)_n = \prod_{k=1}^n (1 - q^k)$. Multiplying this by the character formula for the free boson CFT, with the correct dependence of the energy on the charge $m$ of the vertex operator, gives

$$\text{ch}(q) = q^{\frac{4m^2-1}{16}} \sum_n \frac{q^{\frac{1}{2}n^2 + an + b}}{(q)_n} \prod_l \frac{1}{1 - q^l}. \tag{20}$$

For $a = b = 0$ we get the $V_m$ and $\psi V_m$ sectors,

$$\text{ch}(q) = q^{\frac{4m^2-1}{16}} \frac{\prod_{l>0}(1 + q^{l-1/2})}{\prod_{l>0}(1 - q^l)} = \text{ch}_{V_m}(q) + \text{ch}_{\psi V_m}(q), \tag{21}$$

with

$$\text{ch}_{V_m}(q) = q^{\frac{4m^2-1}{16}}(1 + q + 3q^2 + 5q^3 + \ldots),$$
$$\text{ch}_{\psi V_m}(q) = q^{\frac{4m^2-1}{16}} q^{\frac{1}{2}}(1 + 2q + 4q^2 + \ldots). \tag{22}$$

Choosing $a = \frac{1}{2}$, $b = \frac{1}{16}$ gives the $\sigma V_m$ sector

$$\text{ch}_{\sigma V_m}(q) = q^{\frac{m^2}{4}} \frac{\prod_{l>0}(1 + q^l)}{\prod_{l>0}(1 - q^l)} = q^{\frac{m^2}{4}}(1 + 2q + 4q^2 + 8q^3 + \ldots). \tag{23}$$

In [2] we showed that

1. For open/open BC the $M_2$ spectra correspond to modules $V_m$ and $\psi V_m$, with

$$m = 2f - L - 1/2, \qquad \text{open/open BC.} \tag{24}$$

The module $V_m$ is realised for $f$ even, while $f$ odd leads to $\psi V_m$.

2. For open/$\sigma$ (or $\sigma$/open) BC, the $M_2$ model spectra correspond to modules $\sigma V_m$ with

$$m = 2f - L, \qquad \text{open/}\sigma \text{ BC.} \tag{25}$$

3. For $\sigma/\sigma$ BC we find *both* the modules $V_m$ and $\psi V_m$ at

$$m = 2f - L + 1/2, \qquad \sigma/\sigma \text{ BC.} \tag{26}$$

These findings are consistent with the interpretation that $\sigma$-type BC inject an operator $\sigma V_{1/2}$ into the CFT. The factor $V_{1/2}$ explains the shift in the $m$ values and the fusion rule $\sigma \times \sigma = 1 + \psi$ explains that $\sigma$-type BC on both ends of the chain lead to both the $V_m$ and the $\psi V_m$ modules.

### 3.3  $M_3$ model

In the $M_3$ model we can have a maximum of three particles next to each other on the chain and we therefore have two different constraints available. We can put a constraint (of type $\sigma_1$) forbidding three neighbouring sites to be all occupied ('no 111'), or we can make the constraint stronger (type $\sigma_2$) and forbid two adjacent sites to be both occupied ('no 11').

The CFT for the critical $M_3$ model is a free boson CFT times a $\mathbb{Z}_3$ parafermion CFT, with total central charge $c = 9/5$. The $\mathbb{Z}_3$ parafermions are $\psi_{1,2}$ with $h = 2/3$ and the parafermion spin fields are $\sigma_{1,2}$ with $h = 1/15$ and $\varepsilon$ with $h = 2/5$. The free boson compactification radius is $R = \sqrt{\frac{5}{3}}$. Following the notation in [2], we label the chiral vertex operators as $V_m$,

$$V_m = e^{i\frac{2m}{\sqrt{15}}\varphi}. \tag{27}$$

They have bosonic charge $\tilde{m} = \frac{2m}{3}$ and conformal dimension $h_m = \frac{\tilde{m}^2}{2R^2} = \frac{2m^2}{15}$. The contribution to the energy of a bosonic vertex operator is

$$E_{CFT} = h - \frac{c}{24} = \frac{2m^2}{15} - \frac{3}{40}. \tag{28}$$

The supercharge $\bar{Q}_+$ is the zero-mode of the supercurrent

$$G^+(z) = \psi_1 V_{\frac{5}{2}}(z). \tag{29}$$

The supersymmetric ground states are $|\sigma_{1,2}V_{\pm\frac{1}{4}}\rangle$ and $|V_{\pm\frac{3}{4}}\rangle$. Figure 2 displays the finite-size energies of the states in the various modules.

In figure 3 we plot the numerical $M_3$ model open chain spectra at the critical point for various boundary conditions. It can be seen from the plots that a $\sigma_1$-type BC precisely corresponds to the operator $\sigma_1 V_{1/2}$ and that a $\sigma_2$-type BC corresponds to $\sigma_2 V_1$. Summarising the results (see also figure 3) we find that for open/open BC, the $M_3$ model realises the sectors

$$\begin{array}{ll} V_m & \text{for } f = 0 \mod 3 \\ \psi_1 V_m & \text{for } f = 1 \mod 3 \\ \psi_2 V_m & \text{for } f = 2 \mod 3, \end{array} \tag{30}$$

with $m = \frac{5}{2}f - \frac{3}{2}L - \frac{3}{4}$. For open/$\sigma_1$ BC this becomes

$$\begin{array}{ll} \sigma_1 V_m & \text{for } f = 0 \mod 3 \\ \varepsilon V_m & \text{for } f = 1 \mod 3 \\ \sigma_2 V_m & \text{for } f = 2 \mod 3, \end{array} \tag{31}$$

with $m = \frac{5}{2}f - \frac{3}{2}L - \frac{1}{4}$. This is consistent with the parafermion fusion rules $\sigma_1 \psi_1 = \varepsilon$ and $\sigma_1 \psi_2 = \sigma_2$. For open/$\sigma_2$ BC

$$\begin{array}{ll} \sigma_2 V_m & \text{for } f = 0 \mod 3 \\ \sigma_1 V_m & \text{for } f = 1 \mod 3 \\ \varepsilon V_m & \text{for } f = 2 \mod 3, \end{array} \tag{32}$$

with $m = \frac{5}{2}f - \frac{3}{2}L + \frac{1}{4}$, in agreement with the fusion rules $\sigma_2\psi_1 = \sigma_1$ and $\sigma_2\psi_2 = \varepsilon$.

Putting $\sigma_i$-type BC on both ends, the CFT modules follow the fusion rules $\sigma_1\sigma_1 = \psi_1 + \sigma_2$, $\sigma_1\sigma_2 = 1 + \varepsilon$ and $\sigma_2\sigma_2 = \psi_2 + \sigma_1$.

For the general $M_k$ model, defects eliminating $k + 1 - j$ consecutive '1's will correspond to the $\mathbb{Z}_k$ parafermion spin fields $\sigma_j$, $j = 1, \ldots, k-1$. Upon changing the boundary conditions, the various CFT sectors will shift according to the fusion products with these fields.

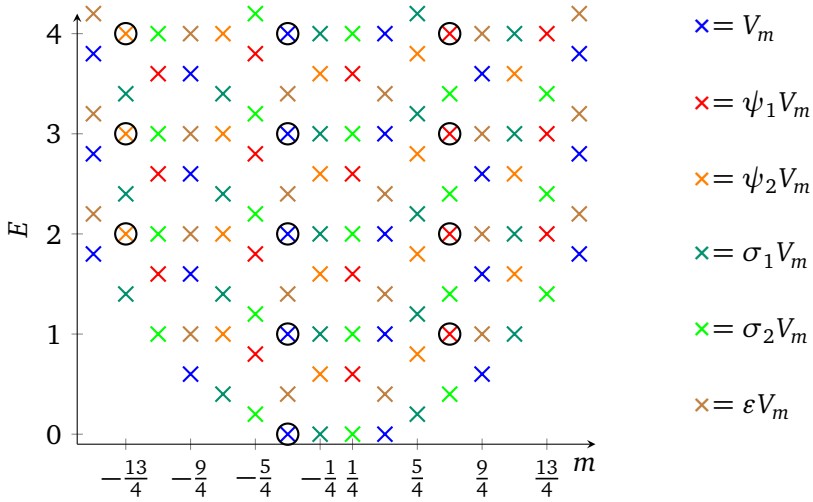

Figure 2: The complete spectrum of the CFT corresponding to the $M_3$ model. The indicated states are for length $L = 5j$ open/open BC.

### 3.3.1 $M_3$ CFT characters

To see that the degeneracies found in the numerical spectra for the $M_3$ model for the different types of boundary conditions are consistent with the CFT we look at the characters of the $\mathbb{Z}_3$ parafermion CFT times a free boson. The Lepowski-Primc formula gives the characters for the $\mathbb{Z}_3$ parafermion part [18]

$$\text{ch}(q) = \sum_{n_1,n_2} \frac{q^{\frac{2}{3}(n_1^2 + n_1 n_2 + n_2^2) + a_1 n_1 + a_2 n_2 + b}}{(q)_{n_1}(q)_{n_2}}. \tag{33}$$

Multiplying this by the partition function for a free boson gives the partition function for the CFT corresponding to the $M_3$ model. For $a_1 = a_2 = b = 0$ we get the $\psi_1 V_m, \psi_2 V_m$ and $V_m$ modules

$$\text{ch}(q) = q^{\frac{2m^2}{15} - \frac{3}{40}} \sum_{n_1,n_2} \frac{q^{\frac{2}{3}(n_1^2 + n_1 n_2 + n_2^2)}}{(q)_{n_1}(q)_{n_2}} \prod_l \frac{1}{1 - q^l}, \tag{34}$$

this gives

$$\text{ch}(q) = q^{\frac{2m^2}{15} - \frac{3}{40}} \left(1 + 2q^{2/3} + q + 4q^{5/3} + 3q^2 + 10q^{8/3} + 6q^3 \right.$$
$$\left. + 18q^{11/3} + 12q^4 + 36q^{14/3} + 21q^5 + \ldots\right). \tag{35}$$

The integer powers of $q$ correspond to the $V_m$ sector. There we thus find degeneracies $1, 1, 3, 6, 12, 21 \ldots$. The fractional powers of $q$ correspond to both the $\psi_1 V_m$ and the $\psi_2 V_m$ modules at the same time. In one of these sectors we thus find the degeneracies $1, 2, 5, 9, 18 \ldots$. This agrees with the numerical spectra of figure 3.

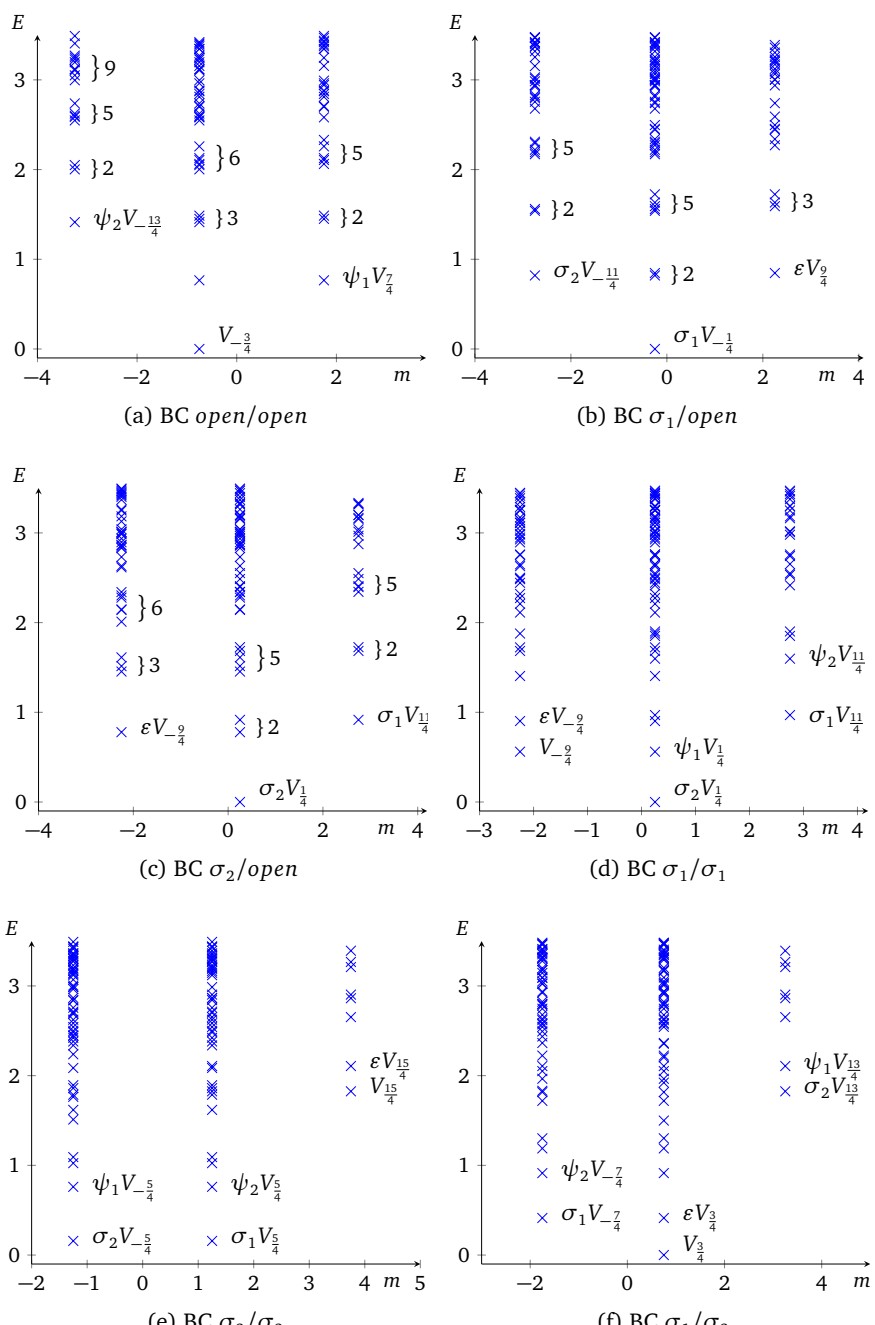

Figure 3: Numerical $M_3$ model spectra with $L = 25$, $f = 14, 15, 16$ up to $E = 3.5$. The labels specify the corresponding CFT modules.

For $a_1 = -\frac{1}{3}, a_2 = -\frac{2}{3}, b = \frac{1}{15}$, eq. (34) gives the $\sigma_1 V_m, \sigma_2 V_m$ and $\varepsilon V_m$ modules

$$
\begin{aligned}
\mathrm{ch}(q) = {}& q^{\frac{2m^2}{15} - \frac{3}{40}} \Big( 2q^{1/15} + q^{2/5} + 4q^{16/15} + 3q^{7/5} + 10q^{31/15} \\
& + 6q^{12/5} + 20q^{46/15} + 13q^{17/5} + 40q^{61/15} + 24q^{22/5} + \dots \Big).
\end{aligned} \tag{36}
$$

In the $\sigma_1 V_m, \sigma_2 V_m$ we find degeneracies $1, 2, 5, 10, 20\dots$ and in the $\varepsilon V_m$ modules we find $1, 3, 6, 13, 24\dots$. The first few of these degeneracies can also be seen in the numerical spectra.

# 4 Continuum limit of the off-critical $M_k$ models

In the appendix we recall that the $\mathcal{N} = 2$ superconformal minimal models, upon perturbation by their least relevant chiral primary field, flow to a massive QFT which in superfield formalism is captured by a superpotential in the form of a Chebyshev polynomial. We expect that, for general $k$, the continuum limit of the integrable staggering perturbation of the $M_k$ lattice model, as given in [15], leads to these same Chebyshev field theories.

## 4.1 $M_1$ model

The continuum limit of the staggered $M_1$ model is the superfield QFT with Chebyshev superpotential

$$W_3(X) = \frac{X^3}{3} - X, \tag{37}$$

where $X$ is the superfield. This theory is equivalent to sine-Gordon theory at its $\mathcal{N} = 2$ supersymmetric point, see section A.3.2 in the appendix.

We remark that the particle structure of the $k = 1$ Chebyshev field theory is similar to that of the $M_1$ model at extreme staggering, with the QFT charge $F$ playing the role of fermion number $f$ in the lattice model. The solitonic particles with charge $F = -1/2$ (called $d_{0,1}$ and $u_{1,0}$ in app. A) correspond to the kinks $K_{0,1}$ and $K_{1,0}$ and the particles with charge $F = 1/2$ ($u_{0,1}$ and $d_{1,0}$) to anti-kinks $\bar{K}_{0,1}$ and $\bar{K}_{1,0}$. The $K_{a,b}$ and $\bar{K}_{a,b}$ form a doublet under $\mathcal{N} = 2$ supersymmetry exactly as in the lattice model. In the field theory the supercharges act on the kinks as given in eq. (140), where $A$ should be read as $K_{a,b}$ and $\bar{A}$ as $\bar{K}_{a,b}$ with $a, b = 0, 1$ or $a, b = 1, 0$.

## 4.2 $M_2$ model

The continuum limit of the staggered $M_2$ model is described by a superfield QFT with Chebyshev superpotential

$$W_4(X) = \frac{X^4}{4} - X^2 + \frac{1}{2}. \tag{38}$$

It is equivalent to $\mathcal{N} = 1$ supersymmetric sine-Gordon theory at the point where there is an additional $\mathcal{N} = 2$ supersymmetry, giving rise to a total of $\mathcal{N} = 3$ left and right supercharges $Q_{L,R}^\pm, Q_{L,R}^0$.

The appearance of $\mathcal{N} = 1$ supersymmetry (which provides a third set of supercharges $Q_{L,R}^0$ in addition to the supercharges for the $\mathcal{N} = 2$ supersymmetry) may be surprising at first. However, a beautiful analysis in [4, 24] showed that the $M_2$ lattice model exhibits a dynamic supersymmetry, with supercharges $Q_0$ and $\bar{Q}_0$ in addition to the manifest $\mathcal{N} = 2$ supersymmetry. These additional lattice supercharges lead to the additional $\mathcal{N} = 1$ supercharges in the continuum limit.

In ref. [4] the perturbing operator was identified as the sum of the fields $\psi\bar{\psi}V_{0,\pm 1}$, which have conformal dimension $h = \bar{h} = 3/4$ (see also appendix A.2.3).

The fundamental particles in this field theory are the kinks $K_{0,\pm}$ and $K_{\pm,0}$, see figure 4. We have given the names in such a way that the kink $K_{a,b}$ has charge $F = -1/2$ and $\bar{K}_{a,b}$ has charge $F = 1/2$. These assignments are consistent with the particle numbers of the (anti-)kinks for the $M_2$ model at extreme staggering, see section 5. In the appendix the structure of the $S$-matrix of the supersymmetric sine-Gordon theory is explained: one part of it is just the sine-Gordon $S$-matrix (of course at its $\mathcal{N} = 2$ supersymmetric point), the other part is the $S$-matrix of the massive tricritical Ising model. The action of the $\mathcal{N} = 3$ supercharges on the kinks is given in eq. (157). We remark that the parity operator that anti-commutes with the $\mathcal{N} = 3$ supercharges exchanges the vacua $|+\rangle$ and $|-\rangle$.

In the M$_2$ lattice model the $\mathcal{N} = 2$ supercharges exchange kinks and anti-kinks without affecting the $\pm$ vacuum structure. We will compare the two situations in section 7.

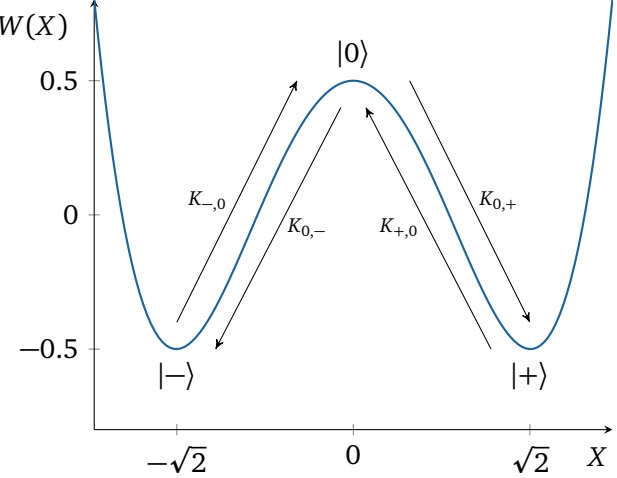

Figure 4: The three supersymmetric vacua and fundamental particles of the Chebyshev superpotential with $k = 2$. The fundamental kinks all have energy $E = \Delta W = 1$. The supercharges relate the kinks $K$ to anti-kinks $\bar{K}$, the latter are not depicted in the figure.

### 4.3 M$_3$ model

We expect that the continuum limit of the staggered M$_3$ model will be the superfield QFT with the number $k + 2 = 5$ Chebyshev superpotential

$$W_5(X) = \frac{X^5}{5} - X^3 + X. \tag{39}$$

This theory has four vacua and we identify the particles with the kinks and anti-kinks in the staggered lattice model, see figure 5.

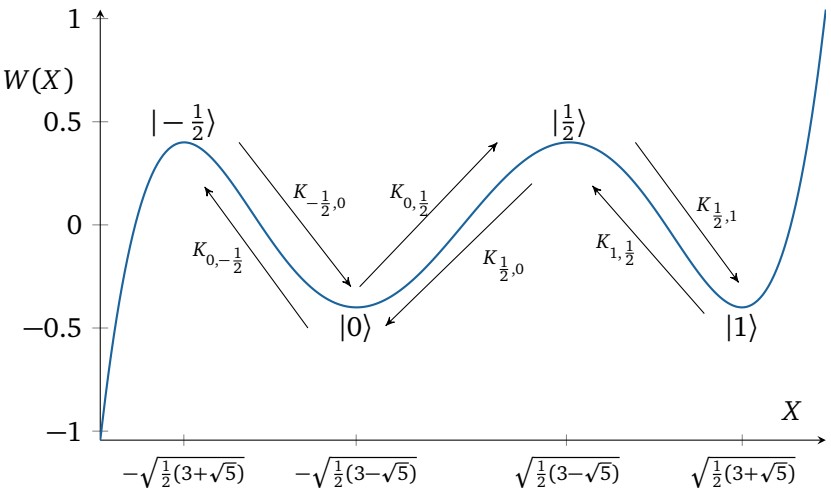

Figure 5: The four supersymmetric vacua and fundamental particles of the Chebyshev superpotential with $k = 3$. The fundamental kinks all have energy $E = \Delta W = 0.8$. The supercharges relate the kinks $K$ to anti-kinks $\bar{K}$, the latter are not depicted in the figure.

# 5  Off-critical $M_k$ models at extreme staggering

We now study the off-critical $M_1$, $M_2$ and $M_3$ models in their so-called extreme staggering limit.

## 5.1  $M_1$ model

For the $M_1$ model with periodic boundary conditions there are two zero-energy ground states at $L = 0 \mod 3$ sites. In the extreme staggering limit $\lambda \to 0$ with staggering $\lambda 11\lambda 11 \ldots$ they take the form

$$|0\rangle = \ldots 100100100\ldots, \quad |1\rangle = \ldots 0(\cdot\cdot)0(\cdot\cdot)0(\cdot\cdot)\ldots \tag{40}$$

where $(\cdot\cdot) = 10 - 01$.

For open chains, these two states may or may not be realised as zero-energy states, depending on BC and on the number of sites. For open BC, staggering $\lambda 11\lambda 11 \ldots$ and $L$ a multiple of 3 the state $|1\rangle$ remains at zero energy but $|0\rangle$ incurs a finite energy. For $L = -1 \mod 3$ the situation is just the opposite. In zero-energy states take the form (with the square brackets indicating open BC)

$$L = 3l, f = l: \quad [0(\cdot\cdot)0\ldots(\cdot\cdot)]$$
$$L = 3l - 1, f = l: \quad [1001\ldots 10]. \tag{41}$$

At extreme staggering and for $L = -2 \mod 3$, the state $|0\rangle$ is a zero-energy state with $f = \frac{L+2}{3}$ while $|1\rangle$ is a zero-energy state with $f = \frac{L-1}{3}$

$$L = 3l - 1, f = l: \quad [1001\ldots 1]$$
$$L = 3l - 2, f = l - 1: \quad [0(\cdot\cdot)0\ldots 0]. \tag{42}$$

As soon as $\lambda > 1$ these two states incur a finite energy and pair up in a supersymmetry doublet.

At extreme staggering, more general eigenstates are formed by connecting the two ground-states with kinks and anti-kinks, which each cost an energy $E = 1$. In our next section we discuss such kinks in the context of the $M_2$ model, where they have a richer structure.

## 5.2  $M_2$ model

In the extreme staggering limit $\lambda \to 0$ of the staggering pattern $1\lambda 1\lambda \ldots$ (see eq. (12)) we find three degenerate, supersymmetric ground states $|-\rangle$, $|0\rangle$, $|+\rangle$ in the $M_2$ model. The excitations are massive kinks that interpolate between any two of them. We use the notation

$$(\cdot 1\cdot) = 110 + 011. \tag{43}$$

For $L = 0 \mod 4$ sites and with periodic BC, the three ground states take the form

$$
\begin{array}{lllllllllll}
\text{staggering} & \lambda & 1 & \lambda & 1 & \lambda & 1 & \lambda & 1 & \lambda & 1 & \ldots \\
|0\rangle = & 1 & 0 & 1 & 0 & 1 & 0 & 1 & 0 & 1 & 0 & \ldots \\
|-\rangle = & 0 & (\cdot & 1 & \cdot) & 0 & (\cdot & 1 & \cdot) & 0 & (\cdot & \ldots \\
|+\rangle = & 1 & \cdot) & 0 & (\cdot & 1 & \cdot) & 0 & (\cdot & 1 & \cdot) & \ldots
\end{array}
\tag{44}
$$

The $|-\rangle$ and $|+\rangle$ ground states are related by a shift over two lattice sites. The state with the 0-s at positions $4l + 1$ is called $|-\rangle$ while $|+\rangle$ has the 0-s at positions $4l + 3$.

We can again investigate which of the three states can be realised as zero-energy states of a finite chain, for a given choice of staggering and BC.

For staggering $1\lambda1\lambda\ldots$ and open BC, one finds that only the state $|+\rangle$ can connect to the left boundary. Similarly, for staggering $\lambda1\lambda1\ldots$ and open BC, the left boundary can accommodate $|0\rangle$ and $|-\rangle$ at zero energy.

Interestingly, all three states $|0\rangle$, $|-\rangle$ and $|+\rangle$ can connect to a left boundary if we choose staggering $\lambda1\lambda1\ldots$ and $\sigma$-BC, imposing the 'no 11' condition on the first two sites. [The same holds true for staggering $1\lambda1\lambda\ldots$ and $\sigma$-type BC with 'no 0' condition.] Arranging for the same situation at the right end, we find that for $L = 4l + 1$, staggering $\lambda1\ldots1\lambda$ and $\sigma$-type 'no 11' BC on both ends we have three zero-energy ground states. In formula (for $L = 13$),

$$
\begin{aligned}
|-\rangle_{\sigma,\sigma} &=_\sigma[\; 0\;(\cdot\;\;1\;\;\cdot)\;0\;(\cdot\;\;1\;\;\cdot)\;0\;(\cdot\;\;1\;\;\cdot)\;0\;]_\sigma \\
|+\rangle_{\sigma,\sigma} &=_\sigma[\; 1\;\;0\;\;0\;(\cdot\;\;1\;\;\cdot)\;0\;(\cdot\;\;1\;\;\cdot)\;0\;\;0\;\;1\;]_\sigma \\
|0\rangle_{\sigma,\sigma} &=_\sigma[\; 1\;\;0\;\;1\;\;0\;\;1\;\;0\;\;1\;\;0\;\;1\;\;0\;\;1\;\;0\;\;1\;]_\sigma
\end{aligned}
\tag{45}
$$

where we denote the $\sigma$-type BC by $\ldots]_\sigma$. Of these states, $|-\rangle_{\sigma,\sigma}$ and $|+\rangle_{\sigma,\sigma}$ have particle number $f = 2l$, while $|0\rangle_{\sigma,\sigma}$ has $f = 2l + 1$.

### 5.2.1 Kinks and anti-kinks

At strong staggering, low energy excitations take the form of (anti-)kinks connecting various ground states. We denote a kink in between ground states $|a\rangle$ and $|b\rangle$, located at site $j$, by $K_{a,b}(j)$. Examples (for staggering type $\lambda1\lambda1\ldots$) are a kink $K_{0,+}$ at site 6

$$
K_{0,+}(6): \qquad [1010\underline{10}0(\cdot1\cdot)0(\cdot1\cdot)0\ldots \tag{46}
$$

or a kink $K_{0,-}$ at site 8

$$
K_{0,-}(8): \qquad [101010\underline{10}0(\cdot1\cdot)0(\cdot1\cdot)0\ldots \tag{47}
$$

where we indicated the location of the kink with an underscore. The supercharge $\bar{Q}_+$ can create an extra particle at the kink location, leading to anti-kinks $\bar{K}_{a,b}(j)$ such as

$$
\begin{aligned}
\bar{K}_{0,+}(6): &\qquad [10101\underline{1}0(\cdot1\cdot)0(\cdot1\cdot)0\ldots \\
\bar{K}_{0,-}(8): &\qquad [1010101\underline{1}0(\cdot1\cdot)0(\cdot1\cdot)0\ldots
\end{aligned}
\tag{48}
$$

The kinks and anti-kinks are superpartners under $Q_+$, $\bar{Q}_+$

$$
\begin{aligned}
\bar{Q}_+ K_{\pm,0}(i) &= \bar{K}_{\pm,0}(i), \quad \bar{Q}_+ K_{0,\pm}(i) = \pm\bar{K}_{0,\pm}(i) \\
Q_+ \bar{K}_{\pm,0}(i) &= K_{\pm,0}(i), \quad Q_+ \bar{K}_{0,\pm}(i) = \pm K_{0,\pm}(i).
\end{aligned}
\tag{49}
$$

It follows that all elementary (anti-)kinks have energy $E = 1$.

### 5.2.2 Multiple (anti-)kinks

In the extreme staggering limit, the spectrum becomes a collection of states with any number of kinks and anti-kinks present. The energy turns out to be additive, there are no bound states of breather-type.

It is import to understand what the minimal spacing of kinks and anti-kinks of given types can be. It turns out that two kinks $K_{\pm,0}(i)$ and $K_{0,\mp}(j)$ can sit at the same location $j = i$. The resulting configurations, of energy $E = 2$, are 'double' kinks $K^{(2)}_{\pm,\mp}$ that connect the $\pm$ to the $\mp$ vacuum,

$$
K^{(2)}_{-,+}(10): \qquad [0(\cdot1\cdot)0(\cdot1\cdot)0\underline{0}0(\cdot1\cdot)0(\cdot1\cdot)0\ldots \tag{50}
$$

| $K_{0,\pm}$ | $1\underline{0}0$ | $E=1$ |
|---|---|---|
| $\bar{K}_{0,\pm}$ | $1\underline{1}0$ | $E=1$ |
| $K_{\pm,0}$ | $0\underline{0}1$ | $E=1$ |
| $\bar{K}_{\pm,0}$ | $0\underline{1}1$ | $E=1$ |
| $K_{\pm,\mp}^{(2)}$ | $0\underline{0}0$ | $E=2$ |
| $K_{\pm,\mp}^{(1,1)}$ | $0\underline{1}0$ | $E=2$ |
| $K_{\pm,\pm}^{(2)}$ | $0(\circ1\circ)0$ | $E=2$ |
| $K_{\pm,\pm}^{(1,1)}$ | $0(\times1\times)0$ | $E=2$ |

Table 1: Elementary (anti-)kinks and some of the kink-(anti-)kink states in the extreme staggering limit of the $M_2$ model.

If we try to bring a kink $K_{\pm,0}$ as close as possible to a second kink $K_{0,\pm}$ or an anti-kink $\bar{K}_{0,\pm}$, we find configurations such as

$$
\begin{aligned}
K_{-,-}^{(2)}(6,8): &\qquad [0(\cdot1\cdot)0(\circ1\circ)0(\cdot1\cdot)\dots \\
K_{-,-}^{(1,1)}(6,8): &\qquad [0(\cdot1\cdot)0(\times1\times)0(\cdot1\cdot)\dots
\end{aligned}
\tag{51}
$$

where $(\circ1\circ)=010$, $(\times1\times)=110-011$. We conclude that the closest approach of $K_{\pm,0}(i)$ and $K_{0,\pm}(j)$ is $j=i+2$, giving the state $K_{\pm,\pm}^{(2)}(i,i+2)$. The superpartner of this state is the linear combination $K_{\pm,\pm}^{(1,1)}(i,i+2)=K_{0,\pm}(i)\bar{K}_{0,\pm}(i+2)+\bar{K}_{0,\pm}(i)K_{0,\pm}(i+2)$. Both these states have energy $E=2$, there is no binding energy. A summary of the elementary kinks and some of the 2-(anti-)kink states is given in table 1.

## 5.3 $M_3$ model

In the extreme staggering limit $\lambda \ll 1$ of the staggering pattern eq. (17), assuming periodic BC on a chain of length $L=5l$, we find the following four ground states [2]

$$
\begin{aligned}
|1\rangle &= \dots 1\underset{\wedge}{1}1001\underset{\wedge}{1}100\dots, &\quad |\tfrac{1}{2}\rangle &= \dots (\cdot\underset{\wedge}{1}\cdots)(\cdot\underset{\wedge}{1}\cdots)\dots, \\
|0\rangle &= \dots 0\underset{\wedge}{1}01101\underset{\wedge}{0}11\dots, &\quad |-\tfrac{1}{2}\rangle &= \dots \underset{\wedge}{0}(\cdot11\cdot)\underset{\wedge}{0}(\cdot11\cdot)\dots,
\end{aligned}
\tag{52}
$$

with $_\wedge$ indicating the position of '$\lambda$' in the staggering pattern eq. (17),
$(\cdot1\cdots)=01101-01110+11001-11010$ and $(\cdot11\cdot)=1110-0111$. All patterns repeat with period 5.

The lattice model kinks have energies given by

$$
E_{a,b}=2|a-b|.
\tag{53}
$$

Remarkable, these kink-energies agree (up to normalisation) with the masses of the fundamental particles in the superfield QFT with the number $k+2=5$ Chebyshev superpotential.

# 6 Counting of $M_2$ model kink states and CFT character formulas

The $M_2$ model spectra are easily tractable in two particular limits. For $\lambda=1$ they organise into finite combinations of modules of the relevant CFT, while for $\lambda=0$ they are understood in terms of states with $n$ kinks and $\bar{n}$ anti-kinks, of energy $E=n+\bar{n}$.

Focusing on open chains, with either open or $\sigma$-type BC at the open ends, we can follow how the $\lambda = 0$ multi-(anti-)kink states connect to states in the CFT spectra upon interpolating $\lambda$ from 0 to 1. In this section our goal is to establish counting formulas for the multi-kink states (at $\lambda = 0$) such that their $q$-deformations correctly reproduce the corresponding contributions to the CFT characters at $\lambda = 1$.

The systematics of the counting procedure are analogous to the counting of quasi-hole excitations over the (fermionic) Moore-Read (MR) state [5]. More generally, a similar connection can be established between the so-called $k$-clustered Read-Rezayi (RR$_k$) states [6] and the staggered M$_k$ models. In our first subsection below we explain this connection. After that we proceed to derive the counting formulas for the M$_2$ model kink states.

## 6.1 Analogy with MR state in thin torus limit

A clear connection between the RR$_k$ quantum Hall states and the M$_k$ models arises in a limit where the many-body states simplify to the point of coming close to states that are in essence product states in an occupation number representation. For the RR$_k$ states this is the so-called thin-torus or Tao-Thouless limit, while for the M$_k$ model a very similar picture arises in the limit of extreme staggering. While both these limits are far from the physical regimes of interest, it has long been understood that the essential structure of the elementary quasi-hole/quasi-particle excitations over fractional quantum Hall states and their (possibly non-Abelian) fusion rules are nicely recovered in the thin-torus limit [7–9]. We will here establish very similar results for the analogous kink/anti-kink excitations of the M$_k$ models, focusing on the case $k = 2$.

In the thin-torus limit the MR states are written as patterns of zeroes and ones where every number corresponds to an orbital in the lowest Landau level (LLL). The orbitals are denoted $\{0, 1, \ldots, N_\phi\}$, where $N_\phi$ is the number of flux quanta, such that the total number of orbitals is $N_{\text{orb}} = N_\phi + 1$. The rule for the MR state is that there should be precisely two particles in any four consecutive orbitals. A violation of the rule, in the form of four consecutive orbitals having only one particle, gives a quasi-hole. As a simple illustration of the thin-torus and extreme staggering limits, we display the patterns of all ground states in periodic BC for $k = 2$. For the MR states these are the six thin-torus groundstates, while for the M$_2$ model these are the three supersymmetric groundstates on a periodic lattice of length $4l$,

$$RR_2 \quad \ldots 11001100\ldots, \quad \ldots 01100110\ldots, \quad \ldots 00110011\ldots, \quad \ldots 10011001\ldots,$$
$$\ldots 10101010\ldots, \quad \ldots 01010101\ldots,$$
$$M_2 \quad \ldots 0(\cdot 1 \cdot)0(\cdot 1 \cdot)0(\cdot 1\ldots, \quad \ldots 1 \cdot)0(\cdot 1 \cdot)0(\cdot 1 \cdot)0\ldots,$$
$$\ldots 10101010101\ldots \tag{54}$$

We now employ the analogy between the CFT, the thin-torus limit of the MR state and the M$_2$ model to learn about open chain BC that open up a two-fold degenerate register. At the level of the CFT, the fundamental degeneracy is that of the two possible fusion channels of the Ising spin-field $\sigma(z)$, that is part of the (chiral) CFT associated to MR and M$_2$ models,

$$\sigma(z)\sigma(w) = (z-w)^{-1/8}[1 + (z-w)^{1/2}\psi(w)] + \ldots. \tag{55}$$

Through the qH-CFT connection, this choice of fusion channel carries over to the fusion product of two fundamental quasi-holes or quasi-particles over the MR state. These excitations, of charge $\pm 1/4$, each carry a single $\sigma$-operator and thus have two choices $1, \psi$ for the fusion channel for any two of them. To see how this plays out in a 'open' geometry, we assume spherical geometry, which we view as an open 'tube' capped by specific boundary conditions at the

two poles. We first inspect the MR ground state in this geometry. Assuming, for definiteness, $N = 8$ particles, the MR ground state requires a number $N_\phi = 2N - 3 = 13$ flux quanta, or $N_{orb} = 14$ LLL orbitals. In thin-torus notation the MR state takes the form

$$|MR, N = 8\rangle = |11001100110011\rangle. \tag{56}$$

The analogous groundstate of the $M_2$ model for $f = 8$ particles on an open chain reads

$$|M_2, f = 8, +\rangle = [(\cdot 1 \cdot)0(\cdot 1 \cdot)0(\cdot 1 \cdot)0(\cdot 1 \cdot)]. \tag{57}$$

Clearly, this needs $L = 15$ sites and staggering pattern $1\lambda 1\lambda \dots \lambda 1$ with $\lambda \to 0$. Note that in neither case is there any sign of degeneracy with other would-be ground states: the ground states are unique and separated from all other states by a gap. The simplest case with two-fold fusion channel degeneracy is that of the MR states with $\Delta N_\phi = 2$, implying the presence of $n = 4$ quasi-holes. The general counting formula for $n$ quasi-holes and a total of $N$ particles reads [20]

$$\# = \sum_{F \equiv N \mod 2} \binom{\frac{N-F}{2} + n}{n} \binom{n/2}{F}. \tag{58}$$

Here the first binomial counts orbital degeneracies of the $n$ quasi-holes, while the second, together with the sum over $F$, pertains to the fusion channel degeneracy. Wishing to view the effects of the quasi-holes as boundary conditions at the two poles, we fix the orbital degeneracy by selecting the states with two quasi-holes at both the north and the south poles, leading to

$$F = 0 \qquad |0110011001100110\rangle$$
$$F = 2 \qquad |1001100110011001\rangle. \tag{59}$$

Returning again to the $M_2$ model, we recognise the corresponding states as

$$|-\rangle_{\sigma,\sigma} = {}_\sigma[0(\cdot 1 \cdot)0(\cdot 1 \cdot)0(\cdot 1 \cdot)0(\cdot 1 \cdot)0]_\sigma$$
$$|+\rangle_{\sigma,\sigma} = {}_\sigma[100(\cdot 1 \cdot)0(\cdot 1 \cdot)0(\cdot 1 \cdot)001]_\sigma \tag{60}$$

at $L = 17$ sites and with staggering $\lambda 1\lambda \dots \lambda 1\lambda$. These are the same states we encountered in eq. (45).

The $\sigma$-type BC that the states eq. (60) require are analogous to the presence of quasi-holes at each pole in the MR state. To understand this we have to compare the open $M_2$ chains with the MR states not on the sphere but on the cylinder. On a cylinder with vacua '1100' at far left and right, we can extend the $F = 0$ state as

$$\dots 1100_{\sigma\sigma}[01100 \dots 110]_{\sigma\sigma}0011 \dots, \tag{61}$$

where $\sigma\sigma$ denotes the two quasi-holes at the boundaries. We can move one of the quasi-holes out from each of the two boundaries to get

$$\dots 1100_\sigma 1010 \dots 10_\sigma[01100 \dots 110]_\sigma 0101 \dots 01_\sigma 0011 \dots. \tag{62}$$

This corresponds to the situation that we have in the $M_2$ model, where $\sigma$-type BC arise from the presence of a single $\sigma$ quantum at a boundary.

## 6.2 Example

As a simple example, let us take an open chain with $L = 4l - 1$ sites, staggering $1\lambda1\ldots$ and open/$\sigma$ BC (meaning 'no 0' condition on the rightmost site $i = 4l - 1$). There is a unique supersymmetric ground state with $f = 2l$ particles,

$$|+\rangle_{o,\sigma} = [(\cdot1\cdot)0\ldots(\cdot1\cdot)0011]_\sigma . \tag{63}$$

This state corresponds to the CFT vacuum $\sigma V_0$ at $E_{\text{CFT}} = L_0 - \frac{1}{16} = 0$. (For the BC used here the CFT charge $m$ is given by $m = 2f - L - 1$.) In addition, these same BC admit 1-kink configurations such as

$$[(\cdot1\cdot)0\ldots(\cdot1\cdot)0\underline{0}101\ldots011]_\sigma . \tag{64}$$

The possible 1-kink states are $K_{+,0}(i)$ with $i = 5, 9, \ldots, 4l - 3$, all with $f = 2l$ particles. These also contribute to the CFT character at $m = 0$. The 1-kink states are counted by a combinatorial factor for choosing one location out of $(l-1)$ possible positions. In the CFT, the lowest value for $E_{\text{CFT}}$ for these 1-kink states turns out to be $E_{\text{CFT}} = 1$, which can be inferred from explicit numerical evaluation. The contribution to the CFT character from 1-kink states is found to be the following $q$-deformation of the kink counting factor

$$\chi_{f=2l}^{4l-1,o,\sigma}[n = 1, \bar{n} = 0] = q\binom{l-1}{1}_q = q(1 + q + q^2 + \ldots + q^{l-2}). \tag{65}$$

Here we use the $q$-binomial which is defined as

$$\binom{n}{m}_q = \frac{(q)_n}{(q)_m(q)_{n-m}} = \prod_{i=0}^{m-1} \frac{1 - q^{n-i}}{1 - q^{i+1}}. \tag{66}$$

We can systematically analyse further contributions $\chi_{f=2l}^{4l-1,o,\sigma}[n, \bar{n}]$ of states with $n$ kinks and $\bar{n}$ anti-kinks to the $m = 0$ character in the CFT, in the form of $q$-deformations of the counting formulas for all multi-(anti-)kink states with $f = 2l$ particles. Some of these are

$$\begin{aligned}
\chi_{f=2l}^{4l-1,o,\sigma}[n = 0, \bar{n} = 0] &= 1 \\
\chi_{f=2l}^{4l-1,o,\sigma}[n = 1, \bar{n} = 0] &= q\binom{l-1}{1}_q = q + q^2 + q^3 + q^4 + q^5 \ldots \\
\chi_{f=2l}^{4l-1,o,\sigma}[n = 1, \bar{n} = 1] &= q\binom{l}{1}_q\binom{l}{1}_q + q^2\binom{l-1}{1}_q\binom{l-1}{1}_q \\
&= q + 3q^2 + 5q^3 + 7q^4 + 9q^5 + \ldots \\
\chi_{f=2l}^{4l-1,o,\sigma}[n = 2, \bar{n} = 1] &= q^3\binom{l}{2}_q\binom{l}{1}_q + q^4\binom{l-1}{2}_q\binom{l-1}{1}_q \\
&= q^3 + 3q^4 + 6q^5 + \ldots \\
\chi_{f=2l}^{4l-1,o,\sigma}[n = 2, \bar{n} = 2] &= q^3\binom{3}{1}_q\binom{l}{2}_q\binom{l}{2}_q + q^6\binom{3}{3}_q\binom{l-1}{2}_q\binom{l-1}{2}_q \\
&= q^3 + 3q^4 + 8q^5 + \ldots
\end{aligned} \tag{67}$$

We derive these expressions in section 6.4 below.

For finite $l$, the sums over all such terms will give a truncation or 'finitisation' of the CFT character $\text{ch}_{\sigma V_0}(q)$. Sending $l$ to infinity then leads to the full CFT character, as given in eq. (23),

$$\lim_{l\to\infty} \sum_{n,\bar{n}} \chi_{f=2l}^{4l-1,o,\sigma}[n, \bar{n}] = 1 + 2q + 4q^2 + 8q^3 + \ldots = \text{ch}_{\sigma V_0}(q). \tag{68}$$

In the sections below we present more general identities of this type.

We refer to [21–23] for other examples where CFT spectra in finitised form are obtained from finite size partition sums of solvable lattice models.

### 6.3 Open/open BC, fusion degeneracies and correspondence to MR quasi-hole state counting

In counting multi-kink states, we encounter a complication due to fusion channel degeneracies. To illustrate this and explain the counting procedure, we zoom in on the case with $L = 4l - 1$, staggering $1\lambda 1 \ldots$ and open BC on both ends. There is again an $f = 2l$ supersymmetric ground state,

$$|+\rangle_{o,o} = [(\cdot 1 \cdot)0\ldots(\cdot 1 \cdot)0(\cdot 1 \cdot)]\,, \tag{69}$$

corresponding to the CFT vacuum $V_{1/2}$ at $E_{\text{CFT}} = 0$. The lowest excited CFT states with $m = 1/2$ are kink/anti-kink states. Before we turn to those, we analyse 2-kink states with $f = 2l - 1$ particles and $m = -3/2$. The two kinks will be of types $K_{+,0}(i)$ and $K_{0,+}(j)$. As explained in section 5, possible choices for $j$ are $j = i + 2, i + 6, \ldots$, while $i = 1, 5, \ldots$, giving $\frac{1}{2}l(l+1)$ two-kink states. We find that the corresponding CFT character is

$$\chi_{f=2l-1}^{4l-1,o,o}[n = 2, \bar{n} = 0] = q\binom{l+1}{2}_q = q(1 + q + 2q^2 + \ldots + q^{2l-2}). \tag{70}$$

Turning to 4-kink states, with $f = 2l - 2$ particles and $m = -7/2$, we realise that there are two choices

$$\text{I} : K_{+,0} K_{0,-} K_{-,0} K_{0,+}, \qquad \text{II} : K_{+,0} K_{0,+} K_{+,0} K_{0,+}. \tag{71}$$

Choice I leads to $\binom{l+3}{4}$ four-kink states while choice II gives $\binom{l+2}{4}$ states. The CFT characters are

$$\chi_{f=2l-2}^{4l-1,o,o}[n = 4, \bar{n} = 0] = q^3\binom{l+3}{4}_q + q^5\binom{l+2}{4}_q. \tag{72}$$

We now observe that the counting of $n$-kink states is identical to the counting of $n$-quasihole excitations over the MR quantum Hall state. Putting $N = 2l - \frac{n}{2}$ in the general counting formula eq. (58) precisely reproduces the counting of the $n$-kink states in the $M_2$ model, as specified above. Indeed, we find that the corresponding CFT characters can be written as (note that for these boundary conditions $n$ is always even)

$$\chi_{f=2l-n/2}^{4l-1,o,o}[n, \bar{n} = 0] = \sum_{F \equiv n/2 \mod 2} q^{\frac{n^2-n}{4} + \frac{F^2}{2}} \binom{\frac{2l - \frac{n}{2} - F}{2} + n}{n}_q \binom{\frac{n}{2}}{F}_q. \tag{73}$$

In eq. (58), the second factor counts the various choices of fusion channels for the $n$ quasi-holes. Adding these numbers gives $2^{\frac{n}{2}-1}$, which is the number of channels opened up by $\frac{n}{2}$ quasi-holes. $F$ counts the number of Majorana fermions associated to the particles in the MR condensate which are not part of a condensed pair. In the CFT, these Majorana's give rise to modes $\psi_{-\frac{1}{2}-j}$, $j = 0, 1, \ldots$ of the fermion $\psi(z)$. Filling the first $F$ of these modes precisely produces the offset energy $\Delta E_{\text{CFT}} = \frac{F^2}{2}$. Despite the similarities we remark that there are differences between the MR and $M_2$ systematics. Most notably, where MR quasi-holes/particles carry charge $\pm\frac{1}{4}$, the $M_2$ model (anti-)kinks have charge $\pm\frac{1}{2}$.

To complete the analysis of the case with open/open BC, we need to extend the counting formula (73) to the more general case where both kinks and anti-kinks are present. Let us first do all cases with $n + \bar{n} = 2$. One quickly finds (again guided by explicit numerics)

$$\chi_{f=2l-1}^{4l-1,o,o}[n = 2, \bar{n} = 0] = q\binom{l+1}{2}_q,$$

$$\chi_{f=2l}^{4l-1,o,o}[n = 1, \bar{n} = 1] = q\binom{l+1}{2}_q + q^2\binom{l}{2}_q = q\binom{l}{1}_q\binom{l}{1}_q, \tag{74}$$

$$\chi_{f=2l+1}^{4l-1,o,o}[n = 0, \bar{n} = 2] = q^2\binom{l}{2}_q.$$

The fine structure in these formulas arises from the fact that the minimal spacing between kink/kink, kink/anti-kink, anti-kink/anti-kink are all different. The structure of these expressions clearly shows the supersymmetric pairing as in eq. (49).

Putting it all together we arrive at the following formula for $L = 4l-1$ sites and open/open BC

$$\chi^{4l-1,o,o}_{f=2l-(n-\bar{n})/2}[n,\bar{n}] =$$
$$\sum_{F \equiv \frac{n+\bar{n}}{2} \mod 2} q^{\frac{F^2}{2}} \binom{\frac{n+\bar{n}}{2}}{F}_q \binom{l+\frac{3n-\bar{n}}{4}-\frac{F}{2}}{n}_q \binom{l+\frac{n+\bar{n}}{4}-\frac{F}{2}}{\bar{n}}_q. \tag{75}$$

We carried out extensive checks and confirmed that the counting formulas agree with numerical evaluation of multiplicities at $\lambda = 0$. For $l$ large, we reproduce the CFT characters eq. (22),

$$\frac{m}{2}-\frac{1}{4} \quad \text{even:} \quad \text{ch}_{V_m}(q) = \lim_{l\to\infty} \sum_{n-\bar{n}=1/2-m} \chi^{4l-1,o,o}_{f=2l+m/2-1/4}[n,\bar{n}]$$
$$\frac{m}{2}-\frac{1}{4} \quad \text{odd:} \quad \text{ch}_{\psi V_m}(q) = \lim_{l\to\infty} \sum_{n-\bar{n}=1/2-m} \chi^{4l-1,o,o}_{f=2l+m/2-1/4}[n,\bar{n}]. \tag{76}$$

We checked these identities, and similar identities given in sections below, by explicit expansion of the $q$-series up to order $q^{15}$.

## 6.4 Open/$\sigma$ BC

Let us now return to the case with open/$\sigma$ BC, we consider only $L = 4l - 1$. Zooming in on 2-kink states, we observe that they can come as

$$\text{I}: K_{+,0} K_{0,+}, \qquad \text{II}: K_{+,0} K_{0,-}. \tag{77}$$

Inspired by the systematics for the open/open case, we associate choice I to $F = 0$ and choice II to $F = 2$ and identify the CFT characters

$$\chi^{4l-1,o,\sigma}_{f=2l-1}[n=2,\bar{n}=0] = q\binom{l+1}{2}_q + q^2\binom{l}{2}_q. \tag{78}$$

We note that, since the $\sigma$-type BC correspond to injecting a $\sigma$ field in the CFT, the CFT Majorana fermion $\psi(z)$ now carries integer modes $\psi_{-j}$, $j = 0, 1, \ldots$. The energy offset for having the first $F$ modes occupied is now $\Delta E_{\text{CFT}} = \frac{F(F-1)}{2}$. The general formula for $n$ kinks, with $n$ even, becomes

$$\chi^{4l-1,o,\sigma}_{f=2l-n/2}[n,\bar{n}=0] = q^{\frac{n^2}{4}} \sum_{F \equiv \frac{n+2}{2} \mod 2} q^{\frac{F(F-1)}{2}} \binom{\frac{n+2}{2}}{F}_q \binom{\frac{2l-\frac{n}{2}-F-1}{2}+n}{n}_q. \tag{79}$$

In a final step we include anti-kinks as well, to arrive at, for $n + \bar{n}$ even,

$$\chi^{4l-1,o,\sigma}_{f=2l-(n-\bar{n})/2}[n,\bar{n}] = q^{\frac{n^2+\bar{n}^2+2\bar{n}}{4}} \times$$
$$\sum_{F \equiv \frac{n+\bar{n}+2}{2} \mod 2} q^{\frac{F(F-1)}{2}} \binom{\frac{n+\bar{n}+2}{2}}{F}_q \binom{l+\frac{3n-\bar{n}}{4}-\frac{F+1}{2}}{n}_q \binom{l+\frac{n+\bar{n}}{4}-\frac{F+1}{2}}{\bar{n}}_q, \tag{80}$$

while for $n + \bar{n}$ odd,

$$\chi^{4l-1,o,\sigma}_{f=2l-(n-\bar{n}-1)/2}[n,\bar{n}] = q^{\frac{n^2+\bar{n}^2+2n+2\bar{n}+1}{4}} \times$$
$$\sum_{F \equiv \frac{n+\bar{n}+1}{2} \mod 2} q^{\frac{F(F-1)}{2}} \binom{\frac{n+\bar{n}+1}{2}}{F}_q \binom{l+\frac{3n-\bar{n}+1}{4}-\frac{F+3}{2}}{n}_q \binom{l+\frac{n+\bar{n}+1}{4}-\frac{F+2}{2}}{\bar{n}}_q. \tag{81}$$

The full CFT characters are recovered as

$$\text{ch}_{\sigma V_m}(q) = \lim_{l \to \infty} \sum_{n,\bar{n}} \chi^{4l-1,o,\sigma}_{f=2l+m/2}[n,\bar{n}] .$$

(82)

## 6.5 $\sigma/\sigma$ BC

We finally turn to the case with $\sigma$-type BC on both ends, we consider only $L = 4l - 2$. In this case, both ends can accommodate each of the three vacua $|+\rangle$, $|-\rangle$, $|0\rangle$, which leads to a larger number of kink-state types. Starting with 1-kink state, we have the choices

$$\text{I} : K_{+,0}, \ K_{0,+}, \qquad \text{II} : K_{-,0}, \ K_{0,-}.$$

(83)

We associate to choice I the value $F = 0$ and to choice II $F = 1$. In addition, the CFT characters have a factor $(1 + q)$ to accommodate for the combinations $K_{a,0} \pm K_{0,a}$. The 1-kink states thus lead to the character

$$\chi^{4l-1,\sigma,\sigma}_{f=2l}[n=1,\bar{n}=0] = q(1+q)\left[\binom{l-1}{1}_q + q^{\frac{1}{2}}\binom{l-2}{1}_q\right].$$

(84)

For these BC, the CFT Majorana fermion $\psi(z)$ again has half-integer modes and we are back to offset energy $\Delta E_{\text{CFT}} = \frac{F^2}{2}$. Note however, that there is no longer a selection rule that links the parity of $F$ to the fermion number $f$. This is because the two $\sigma$-quanta injected by the $\sigma$-type BC fuse according to $\sigma \times \sigma = 1 + \psi$, allowing both parities of the number of CFT quanta $\psi_{-\frac{1}{2}-j}$, regardless of the particle number $f$. For an odd number $n$ of kinks the character formula becomes

$$\chi^{4l-1,\sigma,\sigma}_{f=2l-\frac{(n-1)}{2}}[n,\bar{n}=0] = q^{\frac{n^2+3n}{4}}(1+q)\sum_{F=0,1,\dots} q^{\frac{F^2}{2}}\binom{\frac{n+1}{2}}{F}_q\binom{\lfloor l + \frac{3n-7}{4} - \frac{F}{2}\rfloor}{n}_q.$$

(85)

where we denote by the floor of $x$, written as $\lfloor x \rfloor$, the largest integer not greater than $x$. Allowing for anti-kinks as well but still assuming $n + \bar{n}$ odd, this becomes

$$\chi^{4l-1,\sigma,\sigma}_{f=2l-(n-\bar{n}-1)/2}[n,\bar{n}] = q^{\frac{n^2+\bar{n}^2-2n\bar{n}+3n+3\bar{n}}{4}}(1+q)\times$$
$$\sum_{F=0,1,\dots} q^{\frac{F^2}{2}}\binom{\frac{n+\bar{n}+1}{2}}{F}_q\binom{\lfloor l + \frac{3n-\bar{n}-7}{4} - \frac{F}{2}\rfloor}{n}_q\binom{\lfloor l + \frac{n+\bar{n}-5}{4} - \frac{F}{2}\rfloor}{\bar{n}}_q.$$

(86)

For an even number of kinks we distinguish two situations. For the first, type A, the states at the boundaries are either $|+\rangle$ or $|-\rangle$. For two kinks this leads to

$$\text{A0} : K_{+,0}K_{0,+}, \qquad \text{A1} : K_{-,0}K_{0,+}, \ K_{+,0}K_{0,-}, \qquad \text{A2} : K_{-,0}K_{0,-}.$$

(87)

We associate $F = 0$ to A0, $F = 1$ to A1 and $F = 2$ to A2 and arrive at the character

$$\chi^{4l-1,\sigma,\sigma}_{f=2l-1}[n_A=2,\bar{n}_A=0] = q^{\frac{3}{2}}\left[\binom{l}{2}_q + q^{\frac{1}{2}}\binom{2}{1}_q\binom{l}{2}_q + q^2\binom{l-1}{2}_q\right].$$

(88)

For general even $n_A$ this becomes

$$\chi^{4l-1,\sigma,\sigma}_{f=2l-n_A/2}[n_A,\bar{n}_A=0] = q^{\frac{n_A^2+n_A}{4}}\sum_{F=0,1,\dots} q^{\frac{F^2}{2}}\binom{\frac{n_A+2}{2}}{F}_q\binom{\lfloor l + \frac{3n_A-4}{4} - \frac{F}{2}\rfloor}{n_A}_q.$$

(89)

and including anti-kinks, with $n_A + \bar{n}_A$ even, we find

$$
\chi_{f=2l-(n_A-\bar{n}_A)/2}^{4l-1,\sigma,\sigma}[n_A,\bar{n}_A] = q^{\frac{n_A^2+\bar{n}_A^2+n_A+3\bar{n}_A}{4}} \times
$$
$$
\sum_{F=0,1,\ldots} q^{\frac{F^2}{2}} \binom{\frac{n_A+\bar{n}_A+2}{2}}{F}_q \binom{\lfloor l+\frac{3n_A-\bar{n}_A-4}{4}-\frac{F}{2}\rfloor}{n_A}_q \binom{\lfloor l+\frac{n_A+\bar{n}_A-4}{4}-\frac{F}{2}\rfloor}{\bar{n}_A}_q. \tag{90}
$$

Note that choosing $n_A = \bar{n}_A = 0$ gives the character

$$
\chi_{f=2l}^{4l-1,\sigma,\sigma}[n_A=0,\bar{n}_A=0] = 1 + q^{\frac{1}{2}} . \tag{91}
$$

Clearly, these two states correspond to the $|-\rangle$ and $|+\rangle$ vacua. They are both 0-kink states at $\lambda = 0$ and we see that the correspondence with the CFT states at $\lambda = 1$ is

$$
|-\rangle_{\sigma,\sigma} \longleftrightarrow |V_{-1/2}\rangle, \quad |+\rangle_{\sigma,\sigma} \longleftrightarrow |\psi V_{-1/2}\rangle. \tag{92}
$$

We are left with type B states, which have an even number of kinks and both boundaries in state 0. For two kinks

$$
\text{B0}: K_{0,-}K_{-,0}, \qquad \text{B1}: K_{0,+}K_{+,0}. \tag{93}
$$

We associate $F = 0$ to B0, $F = 1$ to B1 and arrive at the character

$$
\chi_{f=2l}^{4l-1,\sigma,\sigma}[n_B=2,\bar{n}_B=0] = q^4 \left[ \binom{l-1}{2}_q + q^{\frac{1}{2}} \binom{l-2}{2}_q \right]. \tag{94}
$$

For general even $n_B$ this becomes

$$
\chi_{f=2l+1-n_B/2}^{4l-1,\sigma,\sigma}[n_B,\bar{n}_B=0] = q^{\frac{n_B^2+3n_B+2}{4}} \sum_{F=0,1,\ldots} q^{\frac{F^2}{2}} \binom{\frac{n_B}{2}}{F}_q \binom{\lfloor l+\frac{3n_B-10}{4}-\frac{F}{2}\rfloor}{n_B}_q. \tag{95}
$$

and including anti-kinks, with $n_B + \bar{n}_B$ even, we find

$$
\chi_{f=2l+1-(n_B-\bar{n}_B)/2}^{4l-1,\sigma,\sigma}[n_B,\bar{n}_B] = q^{\frac{n_B^2+\bar{n}_B^2+3n_B+5\bar{n}_B+2}{4}} \times
$$
$$
\sum_{F=0,1,\ldots} q^{\frac{F^2}{2}} \binom{\frac{n_B+\bar{n}_B}{2}}{F}_q \binom{\lfloor l+\frac{3n_B-\bar{n}_B-10}{4}-\frac{F}{2}\rfloor}{n_B}_q \binom{\lfloor l+\frac{n_B+\bar{n}_B-6}{4}-\frac{F}{2}\rfloor}{\bar{n}_B}_q. \tag{96}
$$

Choosing $n_B = \bar{n}_B = 0$ gives the character

$$
\chi_{f=2l+1}^{4l-1,\sigma,\sigma}[n_B=0,\bar{n}_B=0] = q^{\frac{1}{2}}, \tag{97}
$$

corresponding to the vacuum $|0\rangle$, so that

$$
|0\rangle_{\sigma,\sigma} \longleftrightarrow |V_{3/2}\rangle. \tag{98}
$$

The CFT character is recovered by summing all contributions

$$
\text{ch}_{V_m}(q) + \text{ch}_{\psi V_m}(q) = \lim_{l\to\infty} \sum_{n,\bar{n}} \chi_{f=2l+m/2+1/4}^{4l-1,\sigma,\sigma}[n,\bar{n}], \tag{99}
$$

where the sum over $n$, $\bar{n}$ includes all three cases $n + \bar{n}$ odd and for $n + \bar{n}$ even types A, B.

# 7 $M_2$ model versus supersymmetric sine-Gordon theory - action of the supercharges

In this section we compare the action of the various supercharges on the kinks in the $M_2$ model and in the supersymmetric sine-Gordon (ssG) field theory.

## 7.1 Kinematics

We consider the $M_2$ model on the infinite open chain, where we denote the location of a kink with a superscript, $K_{a,b}(j) = K_{a,b}^j$. To lowest order in $\lambda$, the lattice model Hamiltonian $H_{M_2}$ acts as

$$H_{M_2}: \quad K_{\pm,0}^m \to K_{\pm,0}^m + \lambda(K_{\pm,0}^{m-4} + K_{\pm,0}^{m+4}) \tag{100}$$

and similar for $K_{0,\pm}^m$ and for the anti-kinks. Constructing plane waves such as

$$K_{0,+}^{(k)} = \sum_l e^{-ikl} K_{0,+}^{4l+2}, \tag{101}$$

we find eigenvalues for energy and momentum

$$E_{M_2} = 1 + 2\lambda \cos k, \quad P_{M_2} = k \tag{102}$$

with the momentum operator defined as the $P = i\log(T_4)$ with $T_4$ the operator that shifts $m \to m+4$. In the supersymmetric sine-Gordon theory the kink states are labelled by the rapidity and we have

$$E_{ssG} = m\cosh(\theta), \quad P_{ssG} = m\sinh(\theta). \tag{103}$$

Clearly, the staggered chain does not have the Poincaré invariance of the supersymmetric sine-Gordon theory (in the latter, this has emerged in the RG flow from the weakly perturbed $M_2$ model towards the fixed point). However, in the long-wavelength limit we can make the comparison, identifying $k$ with $m\theta$.

## 7.2 $M_2$ model supercharges

The paper [4] identified, in addition to the supercharges $Q_+$, $\bar{Q}_+$, additional pairs of what are called dynamical supersymmetries of the $M_2$ lattice model, with charges $Q_-$, $\bar{Q}_-$, and $Q_0$, $\bar{Q}_0$. These supersymmetries change not only the particle number $f$ but also the number $L$ of lattice sites. The operators $Q_-$ and $\bar{Q}_-$ are obtained from $Q_+$ and $\bar{Q}_+$ via conjugation with an operator $S$,

$$Q_- = SQ_+S, \qquad \bar{Q}_- = S\bar{Q}_+S. \tag{104}$$

$S$ represents a $\mathbb{Z}_2$ symmetry which corresponds to 'spin-reversal' in an associated spin-1 XXZ chain. For the infinite open chain the 'spin-reversal' transformation is a good symmetry of the Hamiltonian when $\lambda_{j+2} = \lambda_j$ and $\mu_j = 1/\sqrt{2}$. It leaves invariant the three ground states $|0\rangle$, $|+\rangle$, and $|-\rangle$ and acts on single kinks as

$$S: \quad K_{\pm,0}^m \leftrightarrow \bar{K}_{\pm,0}^m, \qquad K_{0,\pm}^m \to \bar{K}_{0,\mp}^{m+2}, \qquad \bar{K}_{0,\pm}^m \to K_{0,\mp}^{m-2}. \tag{105}$$

We refer to [4] for the definition of $Q_0$ and $\bar{Q}_0$.

We now analyse the action of the $M_2$ model supercharges on (anti-)kinks. The action of the 'manifest' lattice supercharges $Q_+$, $\bar{Q}_+$ is, to zero-th order in $\lambda$, given in eq. (49). Extending this to first order we find

$$\begin{aligned}
\bar{Q}_+ K_{\pm,0}^m &= \bar{K}_{\pm,0}^m + \lambda\bar{K}_{\pm,0}^{m-4} + \dots, & Q_+ \bar{K}_{\pm,0}^m &= K_{\pm,0}^m + \lambda K_{\pm,0}^{m+4} + \dots \\
\bar{Q}_+ K_{0,\pm}^m &= \pm\bar{K}_{0,\pm}^m \pm \lambda\bar{K}_{0,\pm}^{m+4} + \dots, & Q_+ \bar{K}_{0,\pm}^m &= \pm K_{0,\pm}^m \pm \lambda K_{0,\pm}^{m-4} + \dots
\end{aligned} \tag{106}$$

where the $\dots$ indicate terms with multiple kinks.

With eq. (104) and eq. (105) this leads to

$$\begin{aligned}
Q_- K_{\pm,0}^m &= \bar{K}_{\pm,0}^m + \lambda\bar{K}_{\pm,0}^{m-4} + \dots, & \bar{Q}_- \bar{K}_{\pm,0}^m &= K_{\pm,0}^m + \lambda K_{\pm,0}^{m+4} + \dots \\
Q_- K_{0,\pm}^m &= \mp\bar{K}_{0,\pm}^{m+4} \mp \lambda\bar{K}_{0,\pm}^m + \dots, & \bar{Q}_- \bar{K}_{0,\pm}^m &= \mp K_{0,\pm}^{m-4} \mp \lambda K_{0,\pm}^m + \dots
\end{aligned} \tag{107}$$

where the $\dots$ again indicate terms with multiple kinks. It can be checked that this action of $Q_-$, $\bar{Q}_-$ agrees with the action spelled out in eq. (7) of ref [4].

### 7.3 Supercharges in supersymmetric sine-Gordon theory

In appendix A, eq. (157), we specify the action of all supercharges $Q_{L,R}^{0,+,-}$ on the kink states $K_{a,b}(\theta)$ in the supersymmeric sine-Gordon theory.

To write the action on multi-(anti-)kink states, we need to specify the appropriate parity operator. We define a $\mathbb{Z}_2$ operator $\Gamma$, which exchanges the $\pm$ vacua, by

$$
\begin{aligned}
\Gamma: \quad & K_{\pm,0}(\theta) \longleftrightarrow K_{\mp,0}(\theta), \quad \bar{K}_{\pm,0}(\theta) \longleftrightarrow \bar{K}_{\mp,0}(\theta), \\
& K_{0,\pm}(\theta) \longleftrightarrow K_{0,\mp}(\theta), \quad \bar{K}_{0,\pm}(\theta) \longleftrightarrow -\bar{K}_{0,\mp}(\theta).
\end{aligned}
\tag{108}
$$

This operator anti-commutes with all six supercharges and plays the role of the fermion-parity operator for the massive $\mathcal{N} = 3$ superalgebra.

On multi-kink states, the supercharges have the schematic form

$$
Q_{L,R}^a |K^{(1)}(\theta_1)\dots K^{(n)}(\theta_n)\rangle = \sum_{j=1}^n |\Gamma^{(1)}K^{(1)}(\theta_1)\dots\Gamma^{(j-1)}K^{(j-1)}(\theta_{j-1})Q_{L,R}^{(j),a}K^{(j)}(\theta_j)\dots K^{(n)}(\theta_n)\rangle
\tag{109}
$$

with $\Gamma$ as in eq. (108). The lattice model supercharges $Q_\pm$ and $\bar{Q}_\pm$ lack the $\Gamma$-string. They do have an alternative string, extending over all sites $m' < m$ to the left of where the supercharge act, with per site a factor $(-1)^{f_{m'}}$. These lattice model Fermi factors lead to the $\pm$ signs in the action of the lattice model supercharges on kinks of type $K_{0,\pm}$ and $\bar{K}_{0,\pm}$, see eq. (106) and (107). If we wish to express the lattice model supercharges $Q_\pm$ and $\bar{Q}_\pm$ in terms of the supercharges of the supersymmetric sine-Gordon theory, we need to cancel the $\Gamma$-strings. This can be done by taking suitable (even) products.

### 7.4 $M_2$ model vs. supersymmetric sine-Gordon theory

One would expect that the six field theory supercharges $Q_{L,R}^{0,+,-}$ correspond to the six lattice model supercharges $Q_{+,-,0}$ and $\bar{Q}_{+,-,0}$. However, there are clearly a number of subtleties. We already discussed the difference in the dynamical regime (lattice dispersion versus Poincaré invariance) and the difference in the fermion parity operators.

Comparing the field theory supercharges with the lattice model results, we can establish a correspondence, to 1st order in $\lambda$. The precise statement is that, within the supersymmetric sine-Gordon theory, we can define operators $Q_\pm[\text{ssG}]$ which become similar to $Q_\pm[M_2]$, once we identify the degenerate vacua $\{0,+,-\}$ and the corresponding multi-(anti-)kink states between the two theories,

$$
\begin{aligned}
t = -1 \quad : \quad & Q_+[\text{ssG}] = \frac{1}{\sqrt{2}} Q_R^0 (Q_L^- - \lambda Q_R^+) \\
& \bar{Q}_+[\text{ssG}] = -\frac{1}{\sqrt{2}} Q_L^0 (Q_R^- - \lambda Q_L^+) \\
t = 1 \quad : \quad & Q_+[\text{ssG}] = -\frac{1}{\sqrt{2}} Q_L^0 (Q_R^+ + \lambda Q_L^-) \\
& \bar{Q}_+[\text{ssG}] = \frac{1}{\sqrt{2}} Q_R^0 (Q_L^+ + \lambda Q_R^-).
\end{aligned}
\tag{110}
$$

It is instructive to evaluate the anti-commutators of these expressions. For $t = -1$,

$$
\begin{aligned}
\{Q_+[\text{ssG}], \bar{Q}_+[\text{ssG}]\} &= -\frac{1}{2} \{Q_R^0 (Q_L^- - \lambda Q_R^+), Q_L^0 (Q_R^- - \lambda Q_L^+)\} \\
&= \frac{1}{2} Q_R^0 Q_L^0 \left( \{Q_L^-, Q_R^-\} - \lambda \{Q_L^-, Q_L^+\} - \lambda \{Q_R^+, Q_R^-\} \right)
\end{aligned}
$$

$$
\begin{aligned}
&= \frac{1}{2}t(2t - 2\lambda(H - P) - 2\lambda(H + P)) \\
&= t^2 - 2tH \\
&= 1 + 2\lambda\cosh(\theta),
\end{aligned}
\tag{111}
$$

where we used that $Q_R^0 Q_L^0 = Q_L^0 Q_R^0 = t$ and $t = -1$. For $t = 1$,

$$
\begin{aligned}
\{Q_+[\text{ssG}], \bar{Q}_+[\text{ssG}]\} &= -\frac{1}{2}\{Q_L^0(Q_R^+ + \lambda Q_L^-), Q_R^0(Q_L^+ + \lambda Q_R^-)\} \\
&= \frac{1}{2}Q_R^0 Q_L^0\left(\{Q_R^+, Q_L^+\} + \lambda\{Q_L^-, Q_L^+\} + \lambda\{Q_R^+, Q_R^-\}\right) \\
&= \frac{1}{2}t(2t + 2\lambda(H - P) + 2\lambda(H + P)) \\
&= t^2 + 2tH \\
&= 1 + 2\lambda\cosh(\theta).
\end{aligned}
\tag{112}
$$

The two terms in the last line are similar to those in eq. (102). The first, which in the lattice model is related to the kink rest mass, arises in the field theory setting as the square $t^2$ of the topological charge $t$. The second term, of order $\lambda$, is the lattice kink kinetic energy $2\lambda\cos(k)$, which in the supersymmetric sine-Gordon theory takes the relativistic form $2\lambda\cosh(\theta)$. Extending this reasoning to multi-kink states, we see that the contribution from the topological terms in the field theory to the order $\lambda^0$ energy in the lattice model is a contribution of $t^2 = 1$ per kink or anti-kink, in agreement with the lattice model energy operator at $\lambda = 0$.

We can easily extend the correspondence to the lattice model charges $Q_-$ and $\bar{Q}_-$, which take the form

$$
\begin{aligned}
t = -1 \quad : \quad & Q_-[\text{ssG}] = -\frac{1}{\sqrt{2}}Q_R^0(Q_L^+ - \lambda Q_R^-) \\
& \bar{Q}_-[\text{ssG}] = \frac{1}{\sqrt{2}}Q_L^0(Q_R^+ - \lambda Q_L^-) \\
t = 1 \quad : \quad & Q_-[\text{ssG}] = \frac{1}{\sqrt{2}}Q_R^0(Q_R^- + \lambda Q_L^+) \\
& \bar{Q}_-[\text{ssG}] = -\frac{1}{\sqrt{2}}Q_L^0(Q_L^- + \lambda Q_R^+).
\end{aligned}
\tag{113}
$$

From their explicit action on kinks, or from the relation with the field theory supercharges, it becomes clear that the mutual anti-commutators $\{Q_+, Q_-\}$ and $\{\bar{Q}_+, \bar{Q}_-\}$ are non-vanishing, with details depending on the topological charge $t$. This is in contrast to the implementation of these same charges in the $T^4 = 1$ momentum sectors of a finite closed chain, see ref [4].

We refer to [24] for similar results for the lattice model operators $Q_0$ and $\bar{Q}_0$.

# 8  $M_2$ model versus supersymmetric sine-Gordon theory - finite chains

In this section we again compare the kinks in the $M_2$ lattice model with the kinks in the super-symmetric sine-Gordon theory, this time on a finite open chain. The boundaries break some of the supersymmetries and we will not pursue the comparison at the level of the supercharges. Instead, we focus on the kink spectrum. We first (section 8.1) analyse the $M_2$ model kink spectrum on a open chain with $\sigma$-type boundary conditions. We find a fine-structure in the 1-kink spectrum, which has its origin in mixing of kinks of type $K_{0,\pm}$ at the boundary. In section 8.2

we then analyse a similar splitting in the field theory kink spectrum, where the appropriate formalism employs boundary reflection matrices. Comparing the two we see a qualitative agreement.

## 8.1 Mobile $M_2$ model kinks on open chains

We consider an open chain with $L = 4l + 2$ sites, staggering type $\lambda 1\lambda 1\lambda 1 \ldots$ and choose $\sigma$-type boundary conditions with 'no 0' conditions on both the first and the last site. At particle number $f = L/2$ the lowest energy states are 1-kink states of type $K_{0,\pm}$. A total of $l$ kinks $K_{0,-}$ are possible on the sites $i = 4k$ and the same number of kinks $K_{0,+}$ are possible on the sites $i = 4k + 2$ ($k$ integer). Fig. 6 shows the energies (obtained from numerics) of the six 1-kink states at $L = 14, f = 7$.

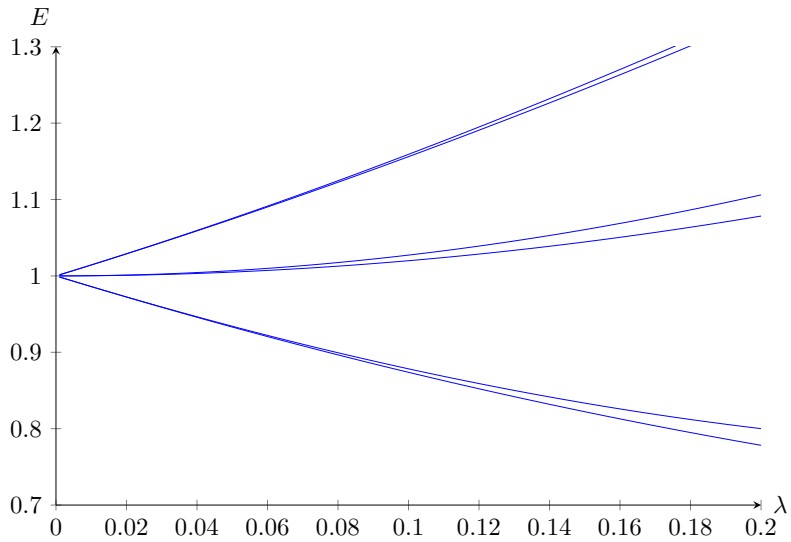

Figure 6: The energies of the six one kink states in the $M_2$ model for small $\lambda$ for $L = 14, f = 7$, $\sigma/\sigma$ BC (no 0/no 0) , staggering $\lambda 1\lambda 1 \ldots \lambda 1$.

Because $\lambda$ is a small parameter we can calculate the energy eigenvalues and the eigenstates of the Hamiltonian perturbatively in $\lambda$. We write $H = H^{(0)} + H^{(1)} + H^{(2)}$, where $H^{(0)}$ does not depend on $\lambda$, $H^{(1)}$ is linear in $\lambda$ and $H^{(2)}$ is quadratic in $\lambda$. We have already seen that the zeroth order of the Hamiltonian just counts the number of kinks. At first order the Hamiltonian of eq. (13) becomes a hopping Hamiltonian for the kinks

$$\langle K_{0,\pm}^{i\pm 4}|H^{(1)}|K_{0,\pm}^i\rangle = \lambda. \tag{114}$$

Hence the 1-kink states have energies

$$E_n = 1 + 2\lambda \cos\frac{n\pi}{l+1}. \tag{115}$$

The $j$-th component of eigenvector number $n$, with $n \in \{1, \ldots, l\}$ has amplitude

$$e_j^{(n)} = \sqrt{\frac{2}{l+1}} \sin\frac{nj\pi}{l+1}. \tag{116}$$

For $n \ll l$ the energy splitting between two of the hopping eigenstates is

$$E_n - E_{n+1} = \frac{2n\pi^2}{l^2}\lambda + \mathcal{O}\left(\frac{n^2}{l^3}\right). \tag{117}$$

The degeneracy between the 1-kink states of types $K_{0,-}$ and $K_{0,+}$ is lifted at second order in $\lambda$. The second order correction to the Hamiltonian acting on the two-dimensional subspaces of $K_{0,-}$ and $K_{0,+}$ hopping eigenstates has two parts,

$$H_{ij}^{(2),\text{tot}} = \sum_{k \neq i,j} \frac{H_{ik}^{(1)} H_{kj}^{(1)}}{E_j - E_k} + H_{ij}^{(2)}, \tag{118}$$

where the sum is over all states $|k\rangle$ with energy different from 1 at order zero. At order $\lambda^2$ the $K_{0,-}$ and $K_{0,+}$ eigenstates do not mix in the bulk, the only term that mixes them comes from the Hamiltonian acting near the boundary. The amplitude for the process that mixes the $n$-th eigenstates of $K_{0,-}$ and $K_{0,+}$ near the boundary comes from the square of the order-$\lambda$ correction to the Hamiltonian and is given by

$$A(l,n) = \left(e_1^{(n)}\right)^2 = \frac{2 \sin^2 \frac{n\pi}{l+1}}{l+1}. \tag{119}$$

The diagonal terms are more complicated because they have many contributions. The total second order correction to the Hamiltonian becomes [24]

$$H_{ij}^{(2),\text{tot}} = \begin{pmatrix} 1 + (l-1/2)A(l,n) & -\frac{1}{\sqrt{2}}A(l,n) \\ -\frac{1}{\sqrt{2}}A(l,n) & 1 + lA(l,n) \end{pmatrix}. \tag{120}$$

This matrix has the eigenvectors $(-1, \sqrt{2})$ and $(\sqrt{2}, 1)$ for all values of $l, n$. The corresponding eigenvalues are $1 + (l-1)A(l,n)$ and $1 + (l+1/2)A(l,n)$. So the energy splitting at order $\lambda^2$ becomes

$$E_n^+ - E_n^- = \frac{3}{2}A(l,n)\lambda^2 = \frac{3n^2\pi^2}{l^3}\lambda^2 + \mathcal{O}\left(\frac{n^3}{l^4}\right). \tag{121}$$

## 8.2 Boundary scattering and kink-spectrum in supersymmetric sine-Gordon theory

We will here compare the result eq. (121) with a similar mixing of kink states in the spectrum of the supersymmetric sine-Gordon theory on a finite segment of length $\mathcal{L}$. To make this match we observe that the for long-wavelength mobile kinks the dispersion eq. (115) of the mobile kinks in the $M_2$ model agrees with the dispersion of long-wavelength kink states in the field theory,

$$E_n^{ssG} = \text{const.} + \frac{1}{2m}p_n^2, \quad p_n = m\theta_n = \frac{\pi n}{\mathcal{L}}, \tag{122}$$

if we identify $\mathcal{L} \to l$ and $2m \to 1/|\lambda|$.

To understand the kink spectrum in the supersymmetric sine-Gordon theory, we need to understand the boundary scattering amplitudes of the kinks. The supersymmetric sine-Gordon kinks can be understood as products of sine-Gordon kinks times kink states in the massive QFT that arises as an integrable perturbation of the tricritical Ising model CFT, see appendix A.2. Their boundary scattering has been analysed in the literature, [25–27], but to our knowledge a complete description of all possible boundary states and the corresponding boundary scattering amplitudes has not been obtained. We will here explore the boundary scattering corresponding to $M_2$ model $\sigma$-type BC at a qualitative level, and argue that it leads to a splitting similar to the result eq. (121) obtained in the $M_2$ model.

The boundary scattering amplitudes for kinks in supersymmetric sine-Gordon theory factorise in a factor corresponding to the sine-Gordon kink/anti-kinks times a factor pertaining to

the perturbed tricritical Ising model. The boundary scattering of the sine-Gordon kink/anti-kinks is necessarily diagonal as the $M_2$ model BC conserve charge and thus prevent processes where kinks reflect into anti-kinks.

What remains is the possibility of mixing of the $\pm$ vacua labelling the single kink states. As in section 8.1 we focus on kinks of type $K_{0,\pm}$, and consider how these reflect off a right boundary with $\sigma$-type BC. An important clue to the identification of their boundary scattering comes from the fact that these BC (in combination with the staggering pattern) allow all three vacua $|+\rangle$, $|-\rangle$ and $|0\rangle$ to live at the boundary at zero energy cost. In the analysis by Nepomechie [25] of boundary scattering in the perturbed tricritical Ising model, a single choice of CFT boundary state was identified, which he calls $(d)$, that allows all three vacua at the boundary. He goes on to analyse the boundary reflection matrices corresponding to this boundary state in the perturbed theory. In addition to diagonal reflection amplitudes $P_\pm(\theta)$ he finds non-zero amplitudes $V_\pm(\theta)$ for processes where $K_{0,+}$ reflects into $K_{0,-}$ or vice versa. The reflection matrix acting on kinks $(K_{0,+}, K_{0,-})$ takes the form

$$R_a^b(\theta) = \begin{pmatrix} P_+ & V_+ \\ V_- & P_- \end{pmatrix} = \begin{pmatrix} 1/\sqrt{2} & -i\sinh(\theta/2) \\ -i\sinh(\theta/2) & 1/\sqrt{2} \end{pmatrix} T(\theta), \tag{123}$$

with $T(\theta)$ an overall diagonal factor. We will proceed on the assumption that this same reflection matrix forms a factor of the boundary scattering amplitudes in the supersymmetric sine-Gordon theory in the situation corresponding to $M_2$ model $\sigma$-type BC.

We can obtain the quantisation of the kink momenta in finite volume $\mathscr{L}$, by demanding that their dynamic phase after propagating back and forth through the system and reflecting off both the right and left boundaries adds up to a multiple of $2\pi$. If the kink starts out moving to the right there is a factor $R_a^b(\theta)$ for the reflection off the right boundary. The reflection off the left boundary gives a scalar $R(\theta)$. Hence we get

$$e^{2i\mathscr{L}p(\theta)} R_a^b(\theta) R(\theta) = 1. \tag{124}$$

The eigenvalues of the normalised reflection matrix are

$$e^{i\phi_\pm} = \lambda_\pm = \frac{1 \pm i\sqrt{2}\sinh(\frac{\theta}{2})}{\sqrt{1 + 2\sinh(\frac{\theta}{2})^2}}, \tag{125}$$

where $\lambda_+$ corresponds to the eigenvector $(1,-1)$ and $\lambda_-$ to the eigenvector $(1,1)$. These are not the same eigenvectors as we found in section 8.1. This is due to the fact that in the $M_2$ model the lattice positions of the kinks differ between $K_{0,+}$ and $K_{0,-}$, which affects the processes near a boundary. This asymmetry is absent in the field theory description.

For small $\theta$ the momentum becomes $p = m\theta$ and the reflection phases can be approximated by $\phi_\pm = \pm\frac{\theta}{\sqrt{2}}$. The quantisation condition becomes

$$e^{2im\theta\mathscr{L}} e^{\pm i\frac{\theta}{\sqrt{2}}} T(\theta) R(\theta) = 1. \tag{126}$$

Writing $T(\theta)R(\theta) = e^{i\phi(\theta)}$ and approximating $\phi(\theta)$ by its value $\phi_0$ at $\theta = 0$, we have

$$2m\theta_n\mathscr{L} \pm \frac{\theta_n}{\sqrt{2}} + \phi_0 = 2\pi n \tag{127}$$

which leads to

$$\theta_n^\pm = \frac{2\pi n - \phi_0}{2m\mathscr{L} \pm \frac{1}{\sqrt{2}}}. \tag{128}$$

Using $E_n^\pm = m \cosh(\theta_n^\pm)$ and expanding in $1/\mathscr{L}$ gives

$$E_n^\pm \approx m + \frac{(2\pi n - \phi_0)^2}{8m\mathscr{L}^2} \mp \frac{(2\pi n - \phi_0)^2}{8\sqrt{2}m^2\mathscr{L}^3} \ . \tag{129}$$

Comparing the leading term to the M$_2$ model dispersion leads to $\phi_0 = 0$, which then gives a fine-structure

$$E_n^+ - E_n^- = -\frac{2\sqrt{2}\pi^2 n^2}{(2m)^2\mathscr{L}^3}. \tag{130}$$

Translating back to the M$_2$ model parameters, we find qualitative agreement with the eq. (121) (up to a multiplicative factor $2\sqrt{2}/3$).

Clearly, the extremely staggered M$_2$ chain differs in its details from the supersymmetric sine-Gordon theory, and we should be careful in making the comparison. Nevertheless, we believe the qualitative comparison is justified and leads to a better understanding of the M$_2$ model in the strongly staggered regime.

# Acknowledgements

We thank Paul Fendley, Liza Huijse, and Rafael Nepomechie for discussions. Part of this work was done at the Rudolf Peierls Institute in Oxford and at the Galileo Galilei Institute in Firenze - we acknowledge the hospitality of these institutions.

**Funding information** TF is supported by the Netherlands Organisation for Scientific Research (NWO). The research is part of the Delta ITP consortium, a program of the Netherlands Organisation for Scientific Research (NWO) that is funded by the Dutch Ministry of Education, Culture and Science (OCW).

# A  Integrable Field Theory

In this appendix some background information is given about the quantum field theories that correspond to the continuum limit of the staggered M$_1$ and M$_2$ models, the sine-Gordon and supersymmetric sine-Gordon models. The M$_1$ model leads to the sine-Gordon theory at the point where it has an $\mathcal{N} = 2$ supersymmetry. The latter is closely related to the supersymmetry in the M$_1$ lattice model. The M$_2$ model leads to what is called the $\mathcal{N} = 1$ supersymmetric sine-Gordon model, at a point where this exhibits an extra $\mathcal{N} = 2$ supersymmetry - we sometimes refer to this as the $\mathcal{N} = 3$ supersymmetric sine-Gordon theory.

## A.1  The sine-Gordon theory

The sine-Gordon theory is described by the action

$$S = \int dt\,dx \left(\frac{1}{8\pi}(\partial_\mu\Phi)^2 - \frac{m^2}{\beta^2}\cos(\beta\Phi)\right), \tag{131}$$

where $\Phi(x,y)$ is a scalar field and $\beta$ is a dimensionless coupling constant. The theory exhibits a discrete symmetry $\Phi \to \Phi + n\frac{2\pi}{\beta}$, $n \in \mathbb{Z}$, which is spontaneously broken at $\beta^2 = 2$. The conformal dimension of $e^{i\beta\Phi(x,y)}$ is $\beta^2$, so the cosine term is exactly marginal for $\beta^2 = 2$. For

$\beta^2 < 2$ the action describes a massive field theory with a particle spectrum which consists of soliton-antisoliton pairs $(A, \bar{A})$ which carry a topological charge

$$T = \frac{\beta}{2\pi} \int_{-\infty}^{\infty} dx \frac{\partial}{\partial x} \Phi(x,y) = \frac{\beta}{2\pi} \left( \Phi(+\infty, y) - \Phi(-\infty, y) \right). \tag{132}$$

In general there are in addition to the solitons also neutral particles in the spectrum. These are the breathers $B_n$, $n = 1, 2, \ldots < \lambda$, where $\lambda$ depends on the value of $\beta$

$$\lambda = \frac{2}{\beta^2} - 1. \tag{133}$$

The scattering of the sine-Gordon solitons is described by [28]

$$\begin{aligned}
A(\theta)A(\theta') &= a(\theta - \theta')A(\theta')A(\theta), \\
\bar{A}(\theta)\bar{A}(\theta') &= a(\theta - \theta')\bar{A}(\theta')\bar{A}(\theta), \\
A(\theta)\bar{A}(\theta') &= b(\theta - \theta')\bar{A}(\theta')A(\theta) + c(\theta - \theta')A(\theta')\bar{A}(\theta) \\
\bar{A}(\theta)A(\theta') &= b(\theta - \theta')A(\theta')\bar{A}(\theta) + c(\theta - \theta')\bar{A}(\theta')A(\theta),
\end{aligned} \tag{134}$$

with

$$\begin{aligned}
a(\theta) &= \sin(\lambda(\pi + i\theta))\rho(-i\theta), \\
b(\theta) &= \sin(-i\lambda\theta)\rho(-i\theta), \\
c(\theta) &= \sin(\lambda\pi)\rho(-i\theta),
\end{aligned} \tag{135}$$

where $\rho(u)$ can be written in terms of gamma-functions.

### A.1.1 Sine-Gordon theory with $\mathcal{N} = 2$ supersymmetry as a perturbed superconformal field theory

We will now start from the $\mathcal{N} = 2$ supersymmetric $c = 1$ CFT and add a perturbing operator which generates a massive sine-Gordon field theory. In this way we find the value of $\beta$ for which the sine-Gordon theory has $\mathcal{N} = 2$ supersymmetry. We thus consider the free boson CFT at the supersymmetric point where the compactification radius is $R = \sqrt{3}$ and add a supersymmetry preserving perturbation known as the Chebyshev perturbation [29, 30]

$$S_{\text{pert}} = g \int d^2z \left( G^-_{R,-1/2} G^-_{L,-1/2} \varphi^+ + G^+_{R,-1/2} G^+_{L,-1/2} \varphi^- \right), \tag{136}$$

where $\varphi^\pm$ are primary fields in the Neveu-Schwarz sector with $h = \bar{h} = 1/6$. This leads to

$$S_{\text{pert}} = 2g \int d^2z \cos(\frac{2}{\sqrt{3}}\Phi). \tag{137}$$

so that adding this term to the action of the free boson gives precisely the sine-Gordon theory with $\beta^2 = 4/3$, $\lambda = 1/2$. We conclude that the sine-Gordon action has an $\mathcal{N} = 2$ supersymmetry at the point $\lambda = \frac{1}{2}$. This is the point that corresponds to the continuum limit of the staggered $M_1$ model. Because $\lambda < 1$ there are no breathers at the $\mathcal{N} = 2$ supersymmetric point.

### A.1.2 Particles in sine-Gordon theory with $\mathcal{N} = 2$ supersymmetry

In the $\beta^2 = 4/3$ sine-Gordon field theory the supercharges $Q^\pm_{L,R}$ satisfy the following algebra [31]

$$\begin{aligned}
\{Q^+_L, Q^-_L\} &= E + P, & \{Q^-_R, Q^+_R\} &= E - P, \\
\{Q^+_L, Q^+_R\} &= \frac{1}{2}(1 - (-1)^T), & \{Q^-_L, Q^-_R\} &= \frac{1}{2}(1 - (-1)^T),
\end{aligned} \tag{138}$$

with all other anti-commutators vanishing. The energy and momentum that enter the algebra are

$$E = m\cosh(\theta), \qquad P = m\sinh(\theta), \qquad (139)$$

and $T$ is the topological charge, see eq. (132).

In the massive theory the $U(1)$ currents $J_L$ and $J_R$ are no longer separately conserved as in the CFT. The combination $F = J_L - J_R$, which is conserved, is identified with the fermion number $F$. This implies that $Q_L^\pm$ has fermion number $F = \pm 1$, $Q_R^\pm$ has $F = \mp 1$. A soliton $A$ has fermion number $F = -1/2$ while $\bar{A}$ has $F = 1/2$. These fractional fermion numbers lead to factors $(-1)^F = \pm i$ when supercharges act on multi-(anti-)soliton states (see [29]).

The action of the supercharges on the (anti-)solitons reads

$$
\begin{aligned}
Q_L^+ A(\theta) + i A(\theta) Q_L^+ &= e^{\theta/2}\bar{A}(\theta), & Q_L^+\bar{A}(\theta) - i\bar{A}(\theta)Q_L^+ &= 0, \\
Q_L^-\bar{A}(\theta) + i\bar{A}(\theta)Q_L^- &= e^{\theta/2}A(\theta), & Q_L^- A(\theta) - i A(\theta)Q_L^- &= 0, \\
Q_R^+\bar{A}(\theta) - i\bar{A}(\theta)Q_R^+ &= e^{-\theta/2}A(\theta), & Q_R^+ A(\theta) + i A(\theta)Q_R^+ &= 0, \\
Q_R^- A(\theta) - i A(\theta)Q_R^- &= e^{-\theta/2}\bar{A}(\theta), & Q_R^-\bar{A}(\theta) + i\bar{A}(\theta)Q_R^- &= 0.
\end{aligned}
\qquad (140)
$$

### A.1.3 Commutation of supercharges with the scattering of solitons

We now explicitly show that the $\mathcal{N} = 2$ supercharges commute with the scattering of the sine-Gordon solitons at the point $\lambda = \frac{1}{2}$. Acting with the supercharge $Q_L^+$ on the left hand side of the first of the scattering relations in eq. (134) gives

$$
\begin{aligned}
Q_L^+ A(\theta)A(\theta') &= e^{\theta/2}\bar{A}(\theta)A(\theta') - i e^{\theta'/2}A(\theta)\bar{A}(\theta') \\
&= \left(b e^{\theta/2} - i c e^{\theta'/2}\right)A(\theta')\bar{A}(\theta) + \left(c e^{\theta/2} - i b e^{\theta'/2}\right)\bar{A}(\theta')A(\theta),
\end{aligned}
\qquad (141)
$$

where $b, c$ depend on $\theta - \theta'$. Acting on the right hand side of the same equation we get

$$
\begin{aligned}
Q_L^+ A(\theta)A(\theta') &= Q_L^+\left(a(\theta - \theta')A(\theta')A(\theta)\right) \\
&= a e^{\theta'/2}\bar{A}(\theta')A(\theta) - i a e^{\theta/2}A(\theta')\bar{A}(\theta).
\end{aligned}
\qquad (142)
$$

Using eq. (135) above it can be verified that these two expressions indeed agree when $\lambda = \frac{1}{2}$. The other case that needs to be checked is the scattering of $A(\theta)$ with $\bar{A}(\theta')$. Here the left hand side gives

$$Q_L^+ A(\theta)\bar{A}(\theta') = a e^{\theta/2}\bar{A}(\theta')\bar{A}(\theta) \qquad (143)$$

while from the right hand side we get

$$Q_L^+ A(\theta)\bar{A}(\theta') = \left(i b e^{\theta/2} + c e^{\theta'/2}\right)\bar{A}(\theta')\bar{A}(\theta). \qquad (144)$$

The two agree if $\lambda = \frac{1}{2}$.

## A.2 Supersymmetric sine-Gordon theory

The $\mathcal{N} = 1$ supersymmetric sine-Gordon theory has the following action (see, for example, [32])

$$S_{ssG} = \int dt\, dx \left(\frac{1}{8\pi}\partial_\mu\Phi\partial^\mu\Phi + i\bar{\Psi}\gamma^\mu\partial_\mu\Psi + m\bar{\Psi}\Psi\cos\left(\frac{\beta}{2}\Phi\right) + \frac{m^2}{4\pi\beta^2}\cos(\beta\Phi)\right), \qquad (145)$$

where $\Phi$ is a real scalar field, $\Psi = (\psi_-, \psi_+)$ a Majorana fermion field, $m$ the mass and $\beta$ the coupling constant. The theory is invariant under $\mathcal{N} = 1$ supersymmetry. The Lagrangian has a

discrete symmetry $\Phi \to \Phi + n\frac{4\pi}{\beta}$, $n \in \mathbb{Z}$. It is also invariant under a half-period shift $\Phi \to \Phi + \frac{2\pi}{\beta}$ if at the same time the relative sign of the fermions is changed $\psi_+ \to -\psi_+, \psi_- \to \psi_-$ (that is $\Psi \to -\gamma^3\Psi$). This can be interpreted as an alternation of the sign of the fermion mass term between consecutive supersymmetric sine-Gordon vacua. At the even vacua $\Phi = 2n\frac{2\pi}{\beta}$ the mass is positive, at the odd vacua $\Phi = (2n+1)\frac{2\pi}{\beta}$ it is negative. When the mass is positive the Majorana fermion describes the high temperature phase of the Ising model and there is only one ground state $|0\rangle$. When the mass is negative it describes the low temperature phase, the $\mathbb{Z}_2$ symmetry is spontaneously broken and there are two ground states $|\pm\rangle$.

### A.2.1 Particles in supersymmetric sine-Gordon theory

The particle content of the supersymmetric sine-Gordon theory is richer than that of the sine-Gordon theory. If a soliton interpolates between an even vacuum and an odd vacuum it can either go from ground state $|0\rangle$ to ground state $|+\rangle$ or to ground state $|-\rangle$, we call these solitons kinks $K_{0,+}$ and $K_{0,-}$ respectively. If a soliton interpolates between an odd and an even vacuum it goes from either $|+\rangle$ or $|-\rangle$ to $|0\rangle$. These are the kinks $K_{+,0}$ and $K_{-,0}$. The antisolitons (anti-kinks) are denoted by a bar. See figure 7 for an overview of all eight particles in supersymmetric sine-Gordon theory.

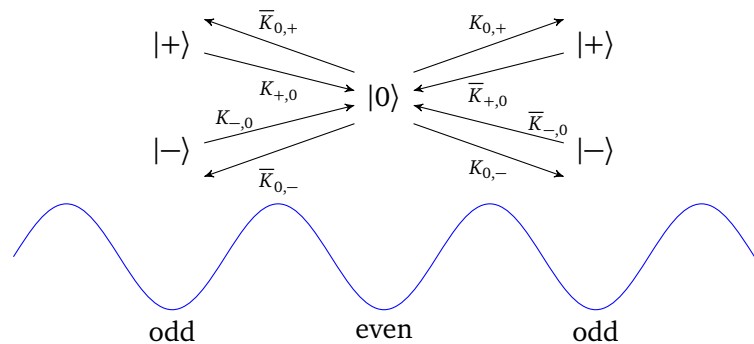

Figure 7: The four kinks and anti-kinks in supersymmetric sine-Gordon theory. Kinks go from left to right, anti-kinks from right to left.

The S-matrix of the supersymmetric sine-Gordon theory decomposes in a part that contains the supersymmetric structure, $S_k$, and a part describing the general sine-Gordon solitons $S_{sG}$ [32],

$$S_{ssG} = S_{sG}(\theta_1 - \theta_2, \lambda) \otimes S_k(\theta_1 - \theta_2). \tag{146}$$

Here $S_{sG}$ is the sine-Gordon S-matrix (see eq. (134)) and

$$\lambda = \frac{2}{\beta^2} - \frac{1}{2}. \tag{147}$$

Note that the definition of $\lambda$ is different from the sine-Gordon case. $S_k$ is equal to the S-matrix of the tricritical Ising model perturbed by the primary field of conformal dimension $h = 3/5$ [33]. The tricritical Ising model CFT is the first in the series of the minimal unitary superconformal models and has central charge $c = 7/10$. This perturbing field should be added with a negative coupling to arrive at a massive field theory with unbroken $\mathcal{N} = 1$ supersymmetry [33]. This theory has three vacua, labeled as $0, \pm1$, which agrees with the supersymmetric vacua described above for the supersymmetric sine-Gordon theory.

The kinks $K_{a,b}, \bar{K}_{a,b}$ can be also be described as consisting of the product of a sine-Gordon soliton $A(\theta)$ or antisoliton $\bar{A}(\theta)$ multiplied by kinks $\mathcal{K}_{a,b}$ between the vacua of the perturbed

tricritical Ising model

$$K_{a,b}(\theta) = A(\theta) \otimes \mathcal{K}_{a,b}(\theta), \qquad \bar{K}_{a,b}(\theta) = \bar{A}(\theta) \otimes \mathcal{K}_{a,b}(\theta) \tag{148}$$

with $a, b = 0, \pm$ (we identify the labels $a = \pm 1$ with $a = \pm$).

The general structure of $S$-matrices in supersymmetric particle theories

$$S(\theta) = S_{BF}(\theta) \otimes S_B(\theta), \tag{149}$$

of which eq. (146) is an example, was given in [34]. The $S$-matrix of the supersymmetric sine-Gordon theory was found in [32, 35, 36].

### A.2.2 $\mathcal{N} = 1$ supersymmetry

In the decomposition described above the $\mathcal{N} = 1$ supersymmetry of supersymmetric sine-Gordon theory originates in the tricritical Ising part of the theory. The $\mathcal{N} = 1$ algebra is [31]

$$(Q_L^0)^2 = E + P, \qquad (Q_R^0)^2 = E - P, \qquad \{Q_L^0, Q_R^0\} = 2t. \tag{150}$$

The $\mathcal{N} = 1$ supersymmetry acts on the $\mathcal{K}_{\pm 1, 0}$ and $\mathcal{K}_{0, \pm 1}$ as [25]

$$\begin{aligned}
Q_L^0 \mathcal{K}_{0,\pm 1}(\theta) - \mathcal{K}_{0,\mp 1}(\theta) Q_L^0 &= \mp e^{\theta/2} \mathcal{K}_{0,\pm 1}, \\
Q_R^0 \mathcal{K}_{0,\pm 1}(\theta) - \mathcal{K}_{0,\mp 1}(\theta) Q_R^0 &= \mp e^{-\theta/2} \mathcal{K}_{0,\pm 1}, \\
Q_L^0 \mathcal{K}_{\pm 1,0}(\theta) - \mathcal{K}_{\mp 1,0}(\theta) Q_L^0 &= \pm i e^{\theta/2} \mathcal{K}_{\mp 1,0}, \\
Q_R^0 \mathcal{K}_{\pm 1,0}(\theta) - \mathcal{K}_{\mp 1,0}(\theta) Q_R^0 &= \mp i e^{-\theta/2} \mathcal{K}_{\mp 1,0}.
\end{aligned} \tag{151}$$

The topological charge, $t$, of $\mathcal{K}_{0,\pm 1}$ is 1, the topological charge of $\mathcal{K}_{\pm 1,0}$ is $-1$. An $n$-kink state is always of the form

$$|\mathcal{K}_{a_1,a_2}, \mathcal{K}_{a_2,a_3}, \mathcal{K}_{a_3,a_4}, \ldots, \mathcal{K}_{a_N, a_{N+1}}\rangle. \tag{152}$$

The total topological charge is given by the sum of the topological charges of the individual kinks and is given by [25]

$$t = -(a_1^2 - a_{N+1}^2). \tag{153}$$

### A.2.3 $\mathcal{N} = 3$ supersymmetric sine-Gordon as a perturbed conformal field theory

The supersymmetric sine-Gordon theory can be seen as a perturbation of the $c = 3/2$ superconformal field theory with perturbation $U = \bar{\Psi}\Psi \cos\frac{\beta\Phi}{2}$ [32]. At the point $\beta^2 = 2$ this perturbation can be written in the form eq. (136), where the Neveu-Schwarz primaries $\varphi_{h,\bar{h}}^{\pm}$ are vertex operators $V_{0,\pm 1}$ with $h = \bar{h} = 1/4$. Indeed, using the explicit form of the supercharges $G_L^{\pm} = \psi V_{\pm 1,\pm 1}$, $G_R^{\pm} = \bar{\psi} V_{\mp 1,\pm 1}$, we have

$$S_{\text{pert}} = g \int d^2 z \, \psi \bar{\psi} \left( V_{0,1} + V_{0,-1} \right) = 2g \int d^2 z \, \psi \bar{\psi} \cos\left(\frac{\Phi}{\sqrt{2}}\right). \tag{154}$$

The form of eq. (136) guarantees that $\mathcal{N} = 2$ supersymmetry is preserved. We conclude that at the point $\beta = \sqrt{2}$ the $\mathcal{N} = 1$ supersymmetry of the supersymmetric sine-Gordon theory is enhanced to an $\mathcal{N} = 3$ supersymmetry. At this point $\lambda = \frac{1}{2}$ and there are no bound states in the theory.

The $\mathcal{N} = 3$ superconformal field theory has an $SU(2)$ symmetry for both right and left movers. The perturbation $S_{\text{pert}}$ does not preserve these separately but does preserve one combination which forms a single $SU(2)$. This combination is given by

$$J^0 = J_L^0 - J_R^0, \quad J^+ = J_L^+ - J_R^-, \quad J^- = J_L^- - J_R^+. \tag{155}$$

Since $J_L^- V_{0,1} = J_R^+ V_{0,-1}$ and $J_R^- V_{0,1} = J_L^+ V_{0,-1}$ it follows that $J^+(V_{0,1} + V_{0,-1}) = J^-(V_{0,1} + V_{0,-1}) = 0$ and thus $S_{\text{pert}}$ is an $SU(2)$ singlet.

### A.2.4 Particles in $\mathcal{N} = 3$ supersymmetric sine-Gordon theory

To write the superalgebra in a uniform form we redefine the $\mathcal{N} = 2$ charges by a factor of $\sqrt{2}$. The massive $\mathcal{N} = 3$ algebra becomes

$$
\begin{aligned}
\{Q_L^+, Q_L^-\} &= 2(Q_L^0)^2 = 2(H + P) \\
\{Q_R^-, Q_R^+\} &= 2(Q_R^0)^2 = 2(H - P) \\
\{Q_L^0, Q_R^0\} &= \{Q_L^-, Q_R^-\} = \{Q_L^+, Q_R^+\} = 2t,
\end{aligned}
\tag{156}
$$

with $t = 1$ for kinks of type $K_{0,\pm}, \bar{K}_{0,\pm}$ and $t = -1$ for $K_{\pm,0}, \bar{K}_{\pm,0}$.

We explicitly write the combined action of the $\mathcal{N} = 3$ supersymmetry on the kinks

$$
\begin{aligned}
Q_L^0 K_{\pm,0}(\theta) - K_{\mp,0}(\theta) Q_L^0 &= \pm i e^{\theta/2} K_{\mp,0}, \\
Q_L^0 \bar{K}_{\pm,0}(\theta) - \bar{K}_{\mp,0}(\theta) Q_L^0 &= \mp i e^{\theta/2} \bar{K}_{\mp,0}, \\
Q_L^+ K_{\pm,0}(\theta) - K_{\mp,0}(\theta) Q_L^+ &= \mp \sqrt{2} i e^{\theta/2} \bar{K}_{\mp,0}, \\
Q_L^+ \bar{K}_{\pm,0}(\theta) - \bar{K}_{\mp,0}(\theta) Q_L^+ &= 0, \\
Q_L^- K_{\pm,0}(\theta) - K_{\mp,0}(\theta) Q_L^- &= 0, \\
Q_L^- \bar{K}_{\pm,0}(\theta) - \bar{K}_{\mp,0}(\theta) Q_L^- &= \mp \sqrt{2} i e^{\theta/2} K_{\mp,0},
\end{aligned}
\tag{157a}
$$

$$
\begin{aligned}
Q_L^0 K_{0,\pm}(\theta) - K_{0,\mp}(\theta) Q_L^0 &= \pm e^{\theta/2} K_{0,\pm}, \\
Q_L^0 \bar{K}_{0,\pm}(\theta) + \bar{K}_{0,\mp}(\theta) Q_L^0 &= \mp e^{\theta/2} \bar{K}_{0,\pm}, \\
Q_L^+ K_{0,\pm}(\theta) - K_{0,\mp}(\theta) Q_L^+ &= -\sqrt{2} e^{\theta/2} \bar{K}_{0,\pm}, \\
Q_L^+ \bar{K}_{0,\pm}(\theta) + \bar{K}_{0,\mp}(\theta) Q_L^+ &= 0, \\
Q_L^- K_{0,\pm}(\theta) - K_{0,\mp}(\theta) Q_L^- &= 0, \\
Q_L^- \bar{K}_{0,\pm}(\theta) + \bar{K}_{0,\mp}(\theta) Q_L^- &= -\sqrt{2} e^{\theta/2} K_{0,\pm},
\end{aligned}
\tag{157b}
$$

$$
\begin{aligned}
Q_R^0 K_{\pm,0}(\theta) - K_{\mp,0}(\theta) Q_R^0 &= \mp i e^{-\theta/2} K_{\mp,0}, \\
Q_R^0 \bar{K}_{\pm,0}(\theta) - \bar{K}_{\mp,0}(\theta) Q_R^0 &= \pm i e^{-\theta/2} \bar{K}_{\mp,0}, \\
Q_R^- K_{\pm,0}(\theta) - K_{\mp,0}(\theta) Q_R^- &= \pm \sqrt{2} i e^{-\theta/2} \bar{K}_{\mp,0}, \\
Q_R^- \bar{K}_{\pm,0}(\theta) - \bar{K}_{\mp,0}(\theta) Q_R^- &= 0, \\
Q_R^+ K_{\pm,0}(\theta) - K_{\mp,0}(\theta) Q_R^+ &= 0, \\
Q_R^+ \bar{K}_{\pm,0}(\theta) - \bar{K}_{\mp,0}(\theta) Q_R^+ &= \pm \sqrt{2} i e^{-\theta/2} K_{\mp,0},
\end{aligned}
\tag{157c}
$$

$$
\begin{aligned}
Q_R^0 K_{0,\pm}(\theta) - K_{0,\mp}(\theta) Q_R^0 &= \pm e^{-\theta/2} K_{0,\pm}, \\
Q_R^0 \bar{K}_{0,\pm}(\theta) + \bar{K}_{0,\mp}(\theta) Q_R^0 &= \mp e^{-\theta/2} \bar{K}_{0,\pm}, \\
Q_R^- K_{0,\pm}(\theta) - K_{0,\mp}(\theta) Q_R^- &= -\sqrt{2} e^{-\theta/2} \bar{K}_{0,\pm}, \\
Q_R^- \bar{K}_{0,\pm}(\theta) + \bar{K}_{0,\mp}(\theta) Q_R^- &= 0, \\
Q_R^+ K_{0,\pm}(\theta) - K_{0,\mp}(\theta) Q_R^+ &= 0, \\
Q_R^+ \bar{K}_{0,\pm}(\theta) + \bar{K}_{0,\mp}(\theta) Q_R^+ &= -\sqrt{2} e^{-\theta/2} K_{0,\pm}.
\end{aligned}
\tag{157d}
$$

The action of the $SU(2)$ currents on the kinks is

$$
\begin{aligned}
J^+ K_{\pm,0} &= \bar{K}_{\pm,0}, & J^+ K_{0,\pm} &= \pm \bar{K}_{0,\pm}, \\
J^- K_{\pm,0} &= 0, & J^- K_{0,\pm} &= 0, \\
J^0 K_{\pm,0} &= -\frac{1}{2} K_{\pm,0}, & J^0 K_{0,\pm} &= -\frac{1}{2} K_{0,\pm}, \\[1ex]
J^+ \bar{K}_{\pm,0} &= 0, & J^+ \bar{K}_{0,\pm} &= 0, \\
J^- \bar{K}_{\pm,0} &= K_{\pm,0}, & J^- \bar{K}_{0,\pm} &= \pm K_{0,\pm}, \\
J^0 \bar{K}_{\pm,0} &= \frac{1}{2} \bar{K}_{\pm,0}, & J^0 \bar{K}_{0,\pm} &= \frac{1}{2} \bar{K}_{0,\pm},
\end{aligned}
\tag{158}
$$

so that $(K_{\pm,0}, \bar{K}_{\pm,0})$ and $(K_{0,\pm}, \pm \bar{K}_{0,\pm})$ form doublets under $SU(2)$.

## A.3  Superfields and superpotentials

We have now seen the field theories that correspond to the continuum limit of the staggered $M_1$ and $M_2$ models. In this section we will turn to the formalism of superfields and superpotentials to be able to easily describe the massive field theories of the higher $M_k$ models. It turns out that this description is also more convenient for understanding the relation of the $M_k$ lattice models with the field theory. A chiral superfield can be written in the form

$$X \sim x + \theta^- \rho + \bar{\theta}^- \eta + \theta^- \bar{\theta}^- \chi, \tag{159}$$

where $x$ is both a left and a right chiral primary, $\rho = G_{L,-1/2}^- x$, $\eta = G_{R,-1/2}^- x$ and $\chi = G_{L,-1/2}^- G_{R,-1/2}^- x$,

$$\bar{X} \sim \bar{x} + \theta^+ \bar{\rho} + \bar{\theta}^+ \bar{\eta} + \theta^+ \bar{\theta}^+ \bar{\chi}, \tag{160}$$

with $\bar{\rho} = G_{L,-1/2}^+ x$, $\bar{\eta} = G_{R,-1/2}^+ x$ and $\bar{\chi} = G_{L,-1/2}^+ G_{R,-1/2}^+ x$.

The $\mathcal{N} = 2$ supersymmetric Landau-Ginzburg action is given in terms of chiral superfields as [37]

$$S = \int d^2z d^4\theta K(X,\bar{X}) + \int d^2z d^2\theta^- W(X) + \int d^2z d^2\theta^+ \bar{W}(\bar{X}), \tag{161}$$

where $d^2\theta^- = d\theta^- d\bar{\theta}^-$ and $d^2\theta^+ = d\theta^+ d\bar{\theta}^+$.

The bosonic part of the superpotential is given by

$$V_{bos} = \left| \frac{\partial W}{\partial X} \right|_{X=x}^2 . \tag{162}$$

For the $k$-th $\mathcal{N} = 2$ superconformal minimal model with $c = (3k)/(k+2)$ the superpotential is $W(X) = X^{k+2}$ [37] .

### A.3.1  Integrable massive field theories with a Chebyshev superpotential

The $\mathcal{N} = 2$ superconformal minimal models perturbed by the least relevant chiral primary field are integrable and the perturbed $k$-th superconformal minimal model is described by the Chebyshev superpotential [29, 30]. We conjecture that these are exactly the field theory descriptions of the $M_k$ models with an integrable staggering.

$$W_{k+2}(X = 2\cos(\theta)) = \frac{2\cos((k+2)\theta)}{k+2}, \tag{163}$$

which gives

$$\begin{aligned} k=1, &\qquad W_3(X) = \frac{X^3}{3} - X \\ k=2, &\qquad W_4(X) = \frac{X^4}{4} - X^2 + \frac{1}{2} \\ k=3, &\qquad W_5(X) = \frac{X^5}{5} - X^3 + X. \end{aligned} \tag{164}$$

The potentials have $k+1$ extrema, given by [30]

$$X^{(r)} = 2\cos\left(\frac{\pi r}{k+2}\right), \quad r = 1,\dots,k+1. \tag{165}$$

The spectrum consists of the $k$ solitons $X_{i,i+1}$ and $k$ antisolitons $X_{i+1,i}$ connecting neighbouring vacua. All these solitons have equal mass [30]

$$m = |\Delta W| = |W_k(X_i) - W_k(X_{i+1})| = \frac{4}{k+2}. \tag{166}$$

Each of these solitons is actually a doublet under supersymmetry, which gives in total $4k$ solitonic particles in the spectrum.

### A.3.2 Comparison of $k = 1$ Chebyshev QFT with sine-Gordon theory

Because the sine-Gordon theory has only bosonic fields in its action we can calculate the bosonic part of the Chebyshev $k = 1$ superpotential and see that it corresponds to the sine-Gordon action. Using $x = V_{0,1}$, the bosonic part gives

$V_{\text{bos}} = |x^2 - \lambda|^2 \sim (V_{0,2} - \lambda)(V_{0,-2} - \lambda) \sim -\lambda \left( V_{0,2} + V_{0,-2} \right)$ which is precisely the perturbation discussed in section A.1.1.

Although the Chebyshev theory with $k = 1$ is in principle the same as the sine-Gordon model at its $\mathcal{N} = 2$ supersymmetric point, the number of solitons appears to be different [30]. In the superfield description we have the soliton $X_{0,1}$ which consists of a doublet $(u_{0,1}, d_{0,1})$ where $u_{0,1}$ has charge $F = +1/2$ and $d_{0,1}$ has charge $F = -1/2$. The corresponding antisolitons $X_{1,0}$ are a doublet $(u_{1,0}, d_{1,0})$ where now $u_{1,0}$ has charge $F = -1/2$ and $d_{1,0}$ has charge $F = 1/2$. The doublet structure occurs because the Dirac equation for the fermion has a zero-energy solution in the presence of the soliton, so the fermion can be either there or not. The relation with the solitons and antisolitons of sine-Gordon $A$ and $\bar{A}$ is non-local. Since the above states are doublets under the supercharges $Q_L^{\pm}$, $Q_R^{\pm}$ whose charges are $F = \pm 1$ and $F = \mp 1$ respectively, we see that we have to identify $u_{0,1}$ and $d_{1,0}$ with $\bar{A}$ and $u_{1,0}$ and $d_{0,1}$ with $A$ [30].

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
