# Peer review of "M$_k$ models: the field theory connection"

_SciPost Physics, doi:SciPost Phys. 3, 004 (2017)_

## Round 1 · Referee Report · Anonymous · 2017-5-10

Strengths

N/A

Weaknesses

N/A

Report

This is an extremely well written paper addressing certain specific examples of the old problem of accurately matching the properties of critical and perturbed lattice models with those of the quantum field theories expected to be relevant in the continuum limit. Only the first three of an infinite series of lattice models are treated, but the analysis is solid and the logic is clearly explained. I expect that the paper will be a very useful resource for future work on these models.

The paper begins with a detailed review of the three lattice models $M_i$, $i=1,2,3$, under consideration, discussing the QFTs that are believed to describe the continuum limits at the critical point as well as certain off-critical regimes. These QFTs are further discussed in the appendix, a very useful addition to the text.

The main results concern a certain type of degeneration of the lattice model when the perturbation is of a specific type called "extreme". Here, the authors discuss the properties of the low-energy states including degenerate vacua and particle-like excitations they call "kinks". These observations are then discussed in the context of QFT.

Readers at all levels will find much of interest here, from theoretical connections between models and theories to numerical analysis of hamiltonian spectra. The methodology appears to be limited to these "small" cases and I did not notice any mention of developing tools to efficiently tackle the models $M_i$ with $i>3$. Nevertheless, the work presented definitely warrants publication and I recommend it strongly.

Requested changes

I have a very short list of minor suggestions, all of which are optional:
- I think the abstract should mention explicitly that the models studied are actually only $M_1$, $M_2$ and $M_3$.
- I had difficulty parsing the definition of the $d_{[a,b],j}^{\dagger}$ before (1). I would change "at position $b$" to "the $b$-th position in the string, $1 \le b \le a$".
- After (4), it is stated that a certain freedom will be addressed in the next section. I believe it is more accurate to say that it will be addressed later in the same subsection, in fact three paragraphs down.
- There is some (standard) sloppiness about identifying CFTs. Sometimes, this is addressed explicitly, eg. in the preamble to Sec. 3 (the N=2 minimal models are "equated" with a boson - parafermion system). Sometimes, it is not, eg. in App. A.1.1, we are told to work with a free boson at $R=\sqrt{3}$ when the notation makes it clear that we are actually working with the (almost equivalent) $N=2$ minimal model of central charge $1$. Wouldn't it actually be better to phrase things in the latter setting so that the appearance of $G$-fields isn't confusing (to non-specialists)? A second example is at the start of App. A.2.3 where we are told to look at the $c=\frac{3}{2}$ $N=2$ minimal model, but are actually working with the (almost equivalent) boson-fermion system.
- This might be naive, but in Sec. 3.1, it is noted that the $M_1$ model with open BCs leads to Ramond modules. Should there not be some mention of the NS sector or why this isn't relevant?
- I appreciate the fact that equations are numbered throughout (it makes referring to things much easier). However, there are equations in Sec. 3.3.1 and App. A.3.1 without numbers.
- I am not sure if I appreciated the "kink" rules in Sec. 5.2. Given the definitions in (41), I would have thought that all the examples in (43--45) would involve $K_{0,-}$ kinks (half are stated to be of $K_{0,+}$-type).
- In (49), it wasn't clear to me what should replace the $\ldots$ so as to obtain ground states. Perhaps the statements could be made less ambiguous?
- In Sec. 6.1 (2nd paragraph), there is a description of a rule for a Moore-Read state and a description of when it is violated. It does not appear to me that the descriptions here actually correspond to a violation of the given rule. Would the authors check please?
- I was a little amused to find, in App. A.1.1, that perturbing the $c=1$ $N=2$ minimal model gives sine-Gordon with $\beta = 2/\sqrt{3}$ and $\beta^2=4/3$...
- In App. A.1.3, after the second unnumbered equation, we read "... indeed give the same when ..." which seems to me to be missing a noun.
- After (138), the term $G^0_{L,R}$ supersymmetry is used. This confused me as I thought maybe it meant the even ($0$) part of some supergroup $G$. It took a while to realise that this was the authors' notation for the $N=1$ superpartner to the EM tensor. I'd advise just deleting $G^0_{L,R}$ as it doesn't seem to be used anywhere else.
- After (146), "form form" should be "form of".

  • validity: top
  • significance: high
  • originality: high
  • clarity: top
  • formatting: perfect
  • grammar: excellent

Author:  Kareljan Schoutens  on 2017-06-21  [id 149]

(in reply to Report 1 on 2017-05-10)
Category:
answer to question
reply to objection
correction

We thank the referee for the constructive remarks, which we used to improve the manuscript (now on arXiv as v2). We addressed all remarks in the revised manuscript.

In the preamble to section 3 we added some remarks on the relation between the boson-parafermion system and the ${\cal N}=2$ supersymmetric CFT and we specified the form of the ${\cal N}=2$ supercurrents. This paragraph now reads

"Throughout this paper we use a description where the $k$-th minimal model of ${\cal N}=2$ supersymmetric CFT is represented as a product of a free boson CFT times a $\mathbb{Z}_k$ parafermion theory. For $k=2$ the parafermion fields are a Majorana fermion $\psi$ and a spin field $\sigma$, while for general $k>1$ we have parafermions $\psi_1, \ldots, \psi_{k-1}$ together with a collection of spin fields. A typical operator in the supersymmetric CFT has a factor originating in the parafermion theory and a factor from the free boson part, the latter taking the form of a vertex operator $V_{p,q}$. Other than in a stand-alone free boson theory, not all charges $p$, $q$ are integers. The ${\cal N}=2$ supercurrents are represented as
\[
G_L^{k,+} = \psi_1 V_{1,{k+2 \over 2k}}, \ \
G_L^{k,-} = \psi_{k-1} V_{-1,-{k+2 \over 2k}}, \ \
G_R^{k,+} = \overline{\psi}_1 V_{1,-{k+2 \over 2k}}, \ \
G_R^{k,-} = \overline{\psi}_{k-1} V_{-1,{k+2 \over 2k}}."
\]
In section 3.1 we added the following remark on the Neveu-Schwarz (NS) sectors

"We remark that Neveu-Schwarz sectors are not compatible with lattice supersymmetry. The Neveu-Schwarz vacuum state in particular gives $E_0=-{c \over 24} < 0$, whereas $E\geq 0$ for all states in the supersymmetric lattice model."

---

## Round 1 · Referee Report · Anonymous · 2017-5-16

Strengths

1. Some new results are presented that are interesting and should be published.

Weaknesses

1. This is a long paper and some sections are not very central to the primary questions. Deleting some material may be worthwhile.

Report

In this article, the authors attempt to establish a dictionary between some lattice models known to possess N=2 supersymmetry (referred to as M_k) and some continuum quantum field theories both at and off their critical points. As the authors say in the introduction, this can be a challenging problem. Even for the Ising model it is reasonably complicated to reconstruct the spin field from the free fermion theory.

Some interesting results are obtained that warrant publication in NPB. For instance, the lattice analog of “kinks” is quite interesting. However I would like to recommend some revisions, which are somewhat minor, before the paper is published.

The discussion and references to the quantum field theory literature is potentially confusing, and in parts plainly inaccurate. To my understanding, the critical point of the M_k models are perturbations of the minimal N=2 supersymmetric models which are a free boson and a Z_k parafermion. The off critical perturbations are by the most relevant operator (Chebyshev). These models are very well understood. The S-matrices, etc. are known. Given this:

1. Reference [33] is known to be wrong. And not in a minor way: the spectrum is wrong, has no kinks, etc. They simply wrote down a manifestly N=2 susy action, but it turned out to be the wrong description. This is obvious from the c=1 case. This was confirmed in ref. [41] where the TBA was studied and confirmed that the S-matrices in [33] had the wrong central charge.
There may be the same problems in [37].

2. The first proposal of ``kinks’’ was in a paper by Zamolodchikov in Int. J. Mod. Phys. It should be cited.

3. The off-critical models that are certainly relevant to this article were first solved in ref. [40], with the correct kink spectrum and S-matrices. It’s the first paper on kinks in the models the authors are studying. The latter reference is only incidentally mentioned at the end of the authors’ article, whereas it is rather central to what they are trying to accomplish.

4. The N=1 supersymmetric sine-Gordon model is discussed extensively in sections 7 and 8, and the appendix. It is not clear why this is relevant since the authors are interested in N=2 models. Some clarification is called for, since these N=1 models have the wrong critical point for the lattice models in question. Perhaps these sections could be deleted. The N=1 susy sine-Gordon model was solved by Ahn, Bernard and LeClair, and also has a kink spectrum. If the authors insist that sections 7,8 are important, some comparisons should be made with this work.

5. The connections with the quantum Hall effect are very tangential. There are some known very interesting CFT’s that are conjectured to be related to QHall physics. Even the Majorana fermion! Subsequently many CFT’s have been applied to QHall physics, and the authors have made significant contributions to this subject. The authors should clarify what is important about the connection between the M_k models and QHall physics, in comparison with the multitude of other CFT’s.
If the connection remains tangential, I recommend that the authors write a separate article aimed at the QHall community.

Requested changes

They are numbered in my report

  • validity: good
  • significance: good
  • originality: good
  • clarity: ok
  • formatting: reasonable
  • grammar: excellent

Author:  Kareljan Schoutens  on 2017-06-21  [id 148]

(in reply to Report 2 on 2017-05-16)
Category:
answer to question
reply to objection
correction

We thank the referee for the thoughtful comments, which we incorporated in v2 of the manuscript.

Points 1-3
We reconsidered and rearranged the references for the background material in Appendix A, taking into account the recommendations by the referee. The paper where Zamolodchikov introduces the kink-picture is the one published in the proc. of the 1989 Beijing workshop - we added the source in ref [33].

Point 4
In section 4.2 we added the following sentence clarifying the appearance of additional ${\cal N}=1$ supercharges in the M$_2$ model and hence in the QFTs describing its continuum limit.

"The appearance of ${\cal N}=1$ supersymmetry (which provides a third set of supercharges $Q^0_{L,R}$ in addition to the supercharges for the ${\cal N}=2$ supersymmetry) may be surprising at first. However, a beautiful analysis in \cite{hagendorf:14, fokkema:16thesis} showed that the M$_2$ lattice model exhibits a dynamic supersymmetry, with supercharges $Q_0$ and $\bar{Q}_0$ in addition to the manifest ${\cal N}=2$ supersymmetry. These additional lattice supercharges lead to the additional ${\cal N}=1$ supercharges in the continuum limit."

We also added references to the original papers on the $S$-matrices for particles in the supersymmetric sine-Gordon theory.

Point 5
The goal of the analysis in section 6 is to understand counting formulas for M$_k$ model kinks in the off-critical regime, and to relate those to the conformal characters that describe the continuum limit. The observation is then that the same CFTs that describe the M$_k$ models at criticality dictate the structure of a class of quantum Hall states, through the celebrated qH-CFT connection. Through this connection, deep results on how to count quasi-particle exitations in quantum Hall systems inspire formulas for the counting of kinks in M$_k$ models at extreme staggering. We insist on keeping the description of the quantum Hall connection in the manuscript, as we know of no other way to derive these counting formulas.

---

## Round 2 · Referee Report · Anonymous (Referee 1) · 2017-6-21

Strengths

n/a

Weaknesses

n/a

Report

No further comments. I believe that the article is ready for publication.

Requested changes

n/a

---

## Round 2 · Author Response

In v2 we have incorporated all comments and suggestions by the two referees. Our response to both referees can be found under Replies to the Referee Reports on the SciPost Submission page for version v1 of our manuscript.

---

## Round 2 · List of Changes

The main changes are the following.

- In the abstract we added that the focus in this paper is on the simplest models, M_1, M_2 and M_3.
- In the preamble to section 3 we added some remarks on the relation between the boson-parafermion system and the N=2 supersymmetric CFT. We also specified the form of the N=2 supercurrents.
- In section 3.1 we added a remark on the Neveu-Schwarz sectors.
- In section 4.2 we added a sentence clarifying the appearance of additional N=1 supercharges in the M_2 model and hence in the QFTs describing its continuum limit.
- We reconsidered and rearranged the references for the background material in Appendix A, taking into account the recommendations by the second referee.
- Also in the Appendix we added references to the original work on the S-matrices for particles in the supersymmetric sine-Gordon theory.
- Following the remarks by the first referee we made several minor corrections.

---

## Editorial Decision

published